# Sediment phosphorus speciation and mobility under dynamic redox conditions

*Chris T. Parsons[*,1], Fereidoun Rezanezhad[1], David W. O'Connell[1,2], Philippe Van Cappellen[1]*

[1]Ecohydrology Research Group and The Wate r Institute, University of Waterloo, 200 University Avenue West, Waterloo, Ontario, Canada.

[2]Department of Civil, Structural and Environmental Engineering, Trinity College Dublin, College Green, Museum Building, Dublin 2, Ireland.

[*]Corresponding author: Chris.Parsons@uwaterloo.ca

KEYWORDS: phosphorus, eutrophication, internal loading, redox cycling, bioturbation

## ABSTRACT

Anthropogenic nutrient enrichment has caused phosphorus (P) accumulation in many freshwater sediments, raising concerns that internal loading from legacy P may delay the recovery of aquatic ecosystems suffering from eutrophication. Benthic recycling of P strongly depends on the redox regime within surficial sediment. In many shallow environments, redox conditions tend to be highly dynamic as a result of, among others, bioturbation by macrofauna, root activity,

sediment resuspension and seasonal variations in bottom water oxygen ($O_2$) concentrations. To
gain insight into the mobility and biogeochemistry of P under fluctuating redox conditions, a
suspension of sediment from a hyper-eutrophic freshwater marsh was exposed to alternating 7-
day periods of purging with air and nitrogen gas ($N_2$), for a total duration of 74 days in a
bioreactor system. We present comprehensive data time series of bulk aqueous and solid phase
chemistry, solid phase phosphorus speciation and hydrolytic enzyme activities demonstrating the
mass balanced redistribution of P in sediment during redox cycling. Aqueous phosphate
concentrations remained low ($\sim$2.5 $\mu$M) under oxic conditions, due to sorption to Iron(III)-
oxyhydroxides. During anoxic periods, once nitrate was depleted, the reductive dissolution of
Iron(III)-oxyhydroxides released P. However, only 4.5% of the released P accumulated in
solution while the rest was redistributed between the $MgCl_2$ and $NaHCO_3$ extractable fractions of
the solid phase. Thus, under the short redox fluctuations imposed in the experiments, P
remobilization to the aqueous phase remained relatively limited. Orthophosphate predominated
at all times during the experiment in both the solid and aqueous phase. Combined P monoesters
and diesters accounted for between 9 and 16% of sediment particulate P. Phosphatase activities
up to 2.4 mmol $h^{-1}$ $kg^{-1}$ indicated the potential for rapid mineralization of organic-P ($P_o$), in
particular during periods of aeration when the activity of phosphomonoesterases was 37% higher
than under $N_2$ sparging. The results emphasize that the magnitude and timing of internal P
loading during periods of anoxia are dependent on both P redistribution within sediments and
bottom water nitrate concentrations.

## Keywords:

$^{31}$P NMR, sequential extractions, coupled biogeochemical cycling, phosphorus, iron, sulfur, redox oscillation, redox fluctuation, bioreactor, internal loading

## INTRODUCTION

It is widely recognized that accelerated eutrophication of freshwater aquatic environments is caused primarily by anthropogenic increases to dissolved phosphorus (P) concentrations in surface water (Smith and Schindler, 2009). Rapid cultural eutrophication of oligo or mesotrophic lacustrine and palustrine systems is often attributed to increased external P loadings originating in agricultural run-off or waste water treatment plant (WWTP) effluent. The resultant excessive algal growth negatively impacts aquatic ecosystems and economic activity (Pretty et al., 2003), as well as increasing the risk of infectious diseases (Chun et al., 2013). Strategies to mitigate eutrophication have aimed to reduce point source and diffuse external phosphorus loadings by instituting agricultural best management practices in the surrounding watershed (McLaughlin and Pike, 2014; Sharpley et al., 1994), limiting P inputs to domestic waste water (Corazza and Tironi, 2011) and upgrading WWTPs (Mallin et al., 2005). However, internal loading of P, from sediments to surface water, remains poorly quantified in many systems, and is often the largest source of error in hydrodynamic and ecological phosphorus models (Kim et al., 2013). Early diagenesis and mineralogical removal of labile autochthonous organic phosphorus ($P_o$) from solution is a complex process and is poorly understood in highly dynamic systems despite exerting a strong influence on the magnitude and timing of internal P loading. This is

particularly true in shallow lakes and wetlands due to the high sediment surface area to water
column depth ratio (Søndergaard et al., 2003).
As policy and infrastructure improvements continue in order to mitigate external P inputs to
aquatic systems, the relative importance of internal P loads from legacy P in sediments to overall
P budgets in eutrophic systems is likely to increase (Reddy et al., 2011).
It has been widely demonstrated through laboratory and field investigations, particularly in
seasonally anoxic lakes, that sustained anoxic conditions induced by water column stratification,
typically result in greater P mobility, and correspondingly higher water column P concentrations
(D. Krom and A. Berner, 1981; Einsele, 1936; Hongve, 1997; Katsev et al., 2006; Mortimer,
1941, 1971; Penn et al., 2000). The microbially mediated reductive dissolution of iron(III)
oxyhydroxides or iron(III) phosphate during sustained periods of anoxia at the sediment water
interface (SWI) has long been considered the main mechanism responsible for P release under
anoxic conditions e.g. (Bonneville et al., 2004; Hyacinthe and Van Cappellen, 2004).
More recently, considerable microbial polyphosphate accumulation and release in response to
alternating oxic-anoxic conditions at the SWI in lacustrine environments has also been shown to
occur (Gächter et al., 1988; Gächter and Meyer, 1993; Hupfer et al., 2007; Sannigrahi and Ingall,
2005). In some cases this accumulation of polyphosphate by the microbial community may
account for 10% of total phosphorus (Hupfer et al., 2007)
However, redox conditions in shallow, heavily bioturbated sediments are more spatially and
temporally variable than in seasonally anoxic lakes (Aller, 1994; Gorham and Boyce, 1989)
resulting in short term redox oscillations even with continuous oxia at the SWI.
Additionally, the coupled biogeochemical cycles of other redox sensitive elements such as sulfur
and carbon have been shown to play important and complex roles in phosphorus mobility
(Gächter and Müller, 2003; Joshi et al., 2015; O'Connell et al., 2015). For example, high bottom
water sulfate concentrations have been shown to increase aqueous P in sediments by decreasing
the permanent mineralogical removal of P within vivianite $[Fe_3(PO_4)_{2(s)}]$ and by decreasing the
abundance of iron(III) oxyhydroxides near to the SWI (Caraco et al., 1989). This is due to the
scavenging of iron during the formation of iron sulfide minerals within sediment during
diagenesis (Gächter and Müller, 2003). Carbon cycling also exerts considerable control over
phosphorus mobility within sediment. The stoichiometry of freshly deposited organic matter
(OM) in eutrophic water bodies approaches that of primary production i.e. $\sim C_{106}:N_{16}:P$ (Berner,
1977). Appreciable P may therefore be released to the aqueous phase when organic carbon is
mineralized during microbial respiration of oxygen, nitrate, ferric iron or sulfate. In addition to
driving N, Fe and S cycling, mineralization of organic carbon and concomitant P release has, in
some places, been shown to be the primary mechanism controlling phosphorus mobility at the
SWI (Joshi et al., 2015).
Core incubations and in-situ flux chambers frequently examine the effects of anoxia on P
mobility from sediments but the effects of repetitive redox oscillations are rarely investigated in
a controlled setting (Frohne et al., 2011; Matisoff et al., 2016; Nürnberg, 1988; Thompson et al.,
2006). Consequently, the cumulative and reversible effects of oxic-anoxic cycling, on P
distribution, speciation and mobility within sediments is poorly understood.
The aim of this study is to elucidate the microbial and geochemical mechanisms of in sediment
phosphorus cycling and release associated with commonly occurring short redox fluctuations
(days) in surficial sediments in shallow eutrophic environments. Particularly, we aimed to: 1)
Quantify the redistribution of P between aqueous and mineral sediment pools during fluctuating
redox conditions 2) Determine if the activity of hydrolytic phosphatase enzymes acting on $P_o$
were influenced by redox conditions, 3) Assess if the proportions of orthophosphate, $P_o$, and
polyphosphate varied systematically with redox conditions and 4) Ascertain if P
mobilization/immobilization mechanisms were reversible or cumulative.
We conducted controlled bioreactor experiments using sediment suspensions, designed to
reproduce cyclic redox conditions analogous to those occurring in nature (Aller, .1994, 2004). A
combination of aqueous chemistry, sediment sequential extractions (Ruttenberg, 1992), $^{31}$P NMR
spectroscopy (Cade-Menun, 2005) and extra-cellular enzyme assays (Deng et al., 2013) were
used to produce a comprehensive dataset describing bulk chemistry, microbial and mineralogical
controls on P mobility and speciation during redox oscillations.
**METHODS**
**Field site and sampling**
Surface sediment (0-12 cm), sediment cores (34 cm long, 10 cm diameter), overlying water and
green filamentous algae (GFA) were collected on September 5, 2013 from West Pond in Cootes
Paradise Marsh (43.26979N, 79.92899W) following established guidelines (U.S. EPA, 9/99).
Cootes Paradise is a hyper-eutrophic, coastal freshwater marsh, which drains into Lake Ontario
via Hamilton Harbour (see Figure 1A-1C). The marsh system suffered severe degradation due to
rapid urbanization, population growth and nutrient loadings in the 20$^{th}$ century (Chow-Fraser et
al., 1998). West Pond in particular received extremely high external P loads from the Dundas
WWTP for several decades prior to the installation of sand filters in 1987 (Painter et al., 1991).
The addition of sand filters, and other improvements, decreased P loadings from the WWTP
from 45 kg P day$^{-1}$ in the early 1970's (Semkin et al., 1976) to 4.5 kg P day$^{-1}$ in the 1980s
(Chow-Fraser et al., 1998) and 2.59 kg P day$^{-1}$ in 2011 (Routledge, 2012). However, high
external P loads resulted in accumulation of legacy P in West Pond sediments with total
phosphorus concentrations reaching 200 $\mu$mol g$^{-1}$ by the 1980's (Theysmeyer et al., 1999).
Consequently, dredging was conducted in 1999 in an attempt to remediate the areas most
affected by growth of green filamentous algae (Bowman and Theysmeyer, 2014)
Despite these restoration efforts and decreases to the external P load, pervasive growth of GFA
during the summer persists in parts of Cootes Paradise (Figure 1A). Cyanobacteria are not
commonly observed at this location, potentially due to the high N:P ratios often associated with
WWTP which utilize tertiary P removal treatment (Conley et al., 2009; Stumm and Morgan,

135 1996).

## Sediment characterization

Sediment cores were sliced every 3 cm, homogenized and characterized with bulk sediment
samples prior to bioreactor experiments. Organic carbon and carbonate depth profiles were
determined by thermo-gravimetric analysis (TGA-Q500, TA Instruments Q500) (Pallasser et al.,
2013). Water content and bulk density ($\rho_b$) of the sliced sediment core were determined
gravimetrically after oven drying (Gardner, 1986). Identification and quantification of crystalline
mineralogy was determined by powder X-ray diffraction (XRD) (Empyrean Diffractometer and
Highscore Plus software Ver. 3.0e PANalytical). The density of benthic macro-invertebrates was
also quantified after sieving two additional 7.5 cm diameter, 18 cm deep cores through 500 $\mu$m
mesh.

## Bioreactor experiment and redox oscillation procedure

An initial, concentrated, sediment suspension of approximately 500 g L$^{-1}$ (dry weight equivalent)
was prepared from freshly sampled sediment (0-12 cm) and filtered overlying water (< 0.45 $\mu$m).
Surface water was used, rather than distilled water, to provide background ionic strength and
avoid osmotic shock to the microbial community.  The concentrated suspension was stirred
vigorously for 5 minutes then passed through a < 500 μm stainless steel sieve to remove larger
solid organic material and macro-invertebrates.   This procedure was repeated until a
homogeneous suspension was achieved. The dry weight was then re-determined, and the sieved
solution was diluted with filtered surface water to a final concentration of $247 \pm 2$ g $L^{-1}$. The
resulting suspension was transferred to a bioreactor system (Applikon Biotechnology) after
Thompson et al (2006) and Parsons et al (2013).  In addition to affording precise temperature
control and continuous logging of temperature, redox potential ($E_h$) and pH, the system offers
significant advancements over previous designs (Thompson 2006, Guo 2007, Parsons 2013). The
$E_h$, pH and dissolved oxygen (DO) were measured using a combined autoclavable Mettler
Toledo InPro 3253i/SG open-junction electrode and an AppliSens Low drift polarographic
sensor.  The InPro electrode system, using a common reference electrode, was chosen to help
avoid potential interference between two electrodes in close proximity. A multi parameter
transmitter was used to display current pH, $E_h$ and temperature, to automatically temperature
correct pH values and to adjust measured $E_h$ to the standard hydrogen electrode (SHE). DO was
calibrated using 100% saturation in air (approximately 0.2905 atm) and 0% saturation in $N_2$ at
constant sparging of 30 ml $min^{-1}$
The suspension was stirred continuously and sparged with 30 mL $min^{-1}$ air for 11 days to
equilibrate prior to the redox oscillation procedure.  During the 11-day oxic equilibration period,
$CO_2$ emissions were monitored in the reactor exhaust gas using an IR sensor (Applikon
Biotechnology).
Subsequently, redox potential ($E_h$) variation was induced by the modulation of sparging gases
(30 ml min$^{-1}$) between $N_2:CO_2$ and $O_2:N_2:CO_2$. The suspension was subjected to five cycles of
anoxia (7 days) and oxia (7 days) at constant temperature (25$^o$C) in the dark, while recording $E_h$,
pH, DO and temperature data. The suspension was sampled on days 1, 3, 5 and 7 of each half-
cycle. To separate solid and aqueous components from the sediment suspension, syringe
extracted samples (15 ml) were centrifuged at 5000 rpm for 20 minutes and the supernatant
filtered through 0.45 μm polypropylene membrane filters prior to all aqueous analysis. For
samples taken during anoxic half cycles, centrifugation, filtering and subsampling were
performed in an anoxic glove box ($N_2:H_2$ 97:3%, $O_2 < 1$ ppmv). Time periods were chosen to be
representative of short temporal fluctuations to redox conditions experienced by surficial
sediments (Aller, 1994; Nikolausz et al., 2008; Parsons et al., 2013). Similarly, the temperature
and dark conditions were chosen to reflect those measured in surficial sediment during summer
months at the field site. Summer conditions were chosen as this is when maximum bioturbation,
microbial activity and OM input are expected within the sediment. Similar, long-running batch
reactor experiments using soil or sediment have previously experienced a slowdown of metabolic
processes due to depletion of labile organic carbon (Parsons et al., 2013). Therefore, gaseous
carbon and nitrogen losses from the reactor were balanced by the addition of 3 g of freeze dried,
ground, GFA to the suspension at the onset of each anoxic cycle. The amount of algae added was
determined based on $CO_2$ production from the reactor during the initial 11 day oxic period.
**Aqueous phase methods**
All reagents used were of analytical grade from Fluka, Sigma-Aldrich or Merck unless stated
otherwise and were prepared with 18.2 MΩ cm$^{-1}$ water (Millipore). Total dissolved Na (70),
K(100), Ca(20), Mg(0.5), Mn(1), Fe(3), Al(100), P (2), Si (15) and S (15) concentrations (MDL
in $\mu$g L$^{-1}$, in brackets) were determined by ICP-OES (Thermo Scientific iCAP 6300) after
filtration (< 0.45 $\mu$m) and acidification with HNO$_3$ to < pH 2. Matrix-matched standards were
used for all calibrations and NIST validated multi-elemental solutions were used as controls.
SRP concentrations were determined by the molybdenum blue/ascorbic acid method on a LaChat
QuickChem 8500 flow injection analyzer system (*4500-P E: Phosphorus by Ascorbic Acid*,
1992; Murphy and Riley, 1962) (MDL = 1.2 $\mu$g P L$^{-1}$). DOC was determined using a Shimadzu
TOC-LCPH/CPN analyzer (Shimadzu) following HCl addition (< pH 2) to degas dissolved
inorganic carbon (DIC) (MDL 71 $\mu$g C L$^{-1}$).
Chloride, nitrate, nitrite and sulfate concentrations were measured by ion chromatography using
a Dionex ICS 5000 equipped with a capillary IonPac® AS18 column. Aqueous sulfide was
stabilized with 20 $\mu$L 1% zinc acetate per mL (Pomeroy, 1954) after filtering and determined by
the Cline method (Cline, 1969) (MDL 0.5 µM). Fe$^{2+}_{(aq)}$ was determined by the ferrozine method
immediately after filtering (Stookey, 1970; Viollier et al., 2000) (MDL 3.8 $\mu$M). All aqueous
analyses were conducted in triplicate. The precision and accuracy for all techniques was < 5
RSD% and ±10% with respect to certified reference materials (where commercially available).
**Solid phase methods: Phosphorus and Iron Speciation**
Phosphorus partitioning within the solid phase in the reactor experiment was evaluated over a
time series by both sequential extractions, using a modification (Baldwin, 1996) of the SEDEX
extraction scheme (Ruttenberg, 1992) and solution $^{31}$P NMR spectroscopy (Cade-Menun, 2005).
The two approaches are complementary; $^{31}$P NMR spectroscopy provides information on the
molecular speciation of phosphorus, while sequential extraction provides information on the
association of the P species with operationally defined solid phase fractions.  Therefore the
combination of these two methods reveals redistribution of P within the solid phase over time
during oxic-anoxic transitions.
The original SEDEX extraction scheme quantifies five different P reservoirs within sediment by
consecutively solubilizing progressively more recalcitrant phases by using extracts of increasing
severity. The reaction mechanisms associated with each extraction step are discussed in detail
within Ruttenberg (1992). A modification of the SEDEX extraction scheme proposed by
Baldwin (1996), used here, incorporates an additional  16 hour, 1M $NaHCO_3$ step ($P_{Hum}$) after the
$P_{Ex}$ step, to differentiate OM associated P which may otherwise be co-extracted during the $P_{Fe}$
step.  The pH of the $NaHCO_3$ extraction step was adjusted to 7.6 to minimize dissolution of
carbonates prior to the $P_{CFA}$ extraction step. A total of 15 samples between day 11 and 74 of the
reactor experiment were analyzed in duplicate by sequential extraction.  A summary of the full
sequential extraction method used, including target phases, reactants, pH, temperature and
reaction times is provided in Table 1.
Changes to iron speciation were also evaluated through a time series during the experiment. To
account for surface-sorbed or freshly precipitated Fe, total $Fe^{2+}$ production during anoxic half-
cycles was estimated by a partial extraction (1 hour, 0.5N HCl) on sampled suspensions.
$Fe^{2+}/Fe^{3+}$ ratios were determined in extracts using a modification of the ferrozine method
(Stookey, 1970; Viollier et al., 2000). Additionally, a thermodynamic model was implemented in
PHREEQC (Parkhurst et al., 1999) to assess the saturation index (SI) of various minerals over
time during the reactor experiment using measured pH, temperature, $E_h$ and concentration data.

## NaOH-EDTA Extraction and Solution [31]P NMR Spectroscopy

Molecular changes to P speciation were evaluated over a time series by solution $^{31}$P NMR. Phosphorus was extracted directly from suspension samples (~2 g dry weight equivalent) prior to $^{31}$P NMR analysis. The method used has been shown to allow quantitative analyses of $P_o$ (monoester and diester), polyphosphates and orthophosphate (Amirbahman et al., 2013; Cade-Menun et al., 2006, p.; Cade-Menun and Preston, 1996; Reitzel et al., 2007; Turner et al., 2003b). Briefly, Samples were extracted in 25 mL of 0.25 M (NaOH) and $Na_2EDTA$ (0.05 M) at ambient laboratory temperature (~22 $^o$C) for 4 hours. Subsequently, the tubes were centrifuged (3500 rpm for 20 minutes), the supernatant extracted via syringe then neutralized with 2M HCl to a pH of 7 to avoid the breakdown of polyphosphates during freeze drying (Cade-Menun et al., 2006). This solution was then filtered to < 0.45 μm. Prior to freeze-drying 1 mL aliquots of each sample were diluted and analyzed by ICP-OES spectroscopy for Al, Ca, Fe, Mg, Mn and P. The remaining extracts were frozen at -80 $^o$C and lyophilized for 48 hours. The lyophilized extracts were re-dissolved in 1.0 ml $D_2O$, 0.6 ml 10 M NaOH, and 0.6 ml of the NaOH-EDTA extractant solution and were allowed to stand for 10 min with occasional vortexing. Samples were centrifuged for 20 min at 3500 rpm, transferred to 10-mm NMR tubes, and stored at 4 °C before analysis within 12 hours.

Solution $^{31}$P NMR spectra were obtained using a 600-MHz spectrometer equipped with a 10-mm broadband probe. The NMR parameters were: 90° pulse, 0.68-s acquisition time, 4.32-s pulse delay, 12 Hz spinning, 20 °C, 2200 to 2900 scans (3-4h) for 0-5cm sediment samples (Cade-Menun et al., 2010). Phosphorus compounds were identified by their chemical shifts related to an external orthophosphoric acid standard, with the orthophosphate peak in all spectra standardized to 6ppm. Peak areas were calculated by integration on spectra processed with 10 and 7-Hz line

broadening, using NUTS software (Acorn NMR, Livermore CA, 2000 edition). Peak
assignments were grouped into compounds or groups of specific compound classes if direct
identifications could not be made (Cade-Menun, 2005).

## Extracellular enzyme assays

Rates of enzymatic hydrolysis of $P_o$ were estimated through extracellular enzyme activities
(EEA). Degradation rates for phosphomonoesters, phosphodiesters and pyrophosphate were
determined fluorometrically through use of the MUF tagged substrates; MUF phosphate (MUP),
Bis(MUF)phosphate (DiMUP, Chem-Impex International), and MUF pyrophosphate, (PYRO-P),
Chem-Impex International) respectively. Additionally MUF β-D-glucopyranoside (MUGb) was
used in order to compare phosphatase enzyme activity to the activity of β-glucosidase (cellulase)
(Dunn et al., 2013). Enzyme activities were determined using a microplate reader (Flexstation3,
Molecular Devices) using a modification of Deng et al (2013). Briefly, 1 g dry weight
equivalent of suspension from the reactor was stirred with 100 mL of 100 mM HEPES buffer at
pH 7.5 in a pyrex dish for 10 minutes at 280 rpm to allow for complete homogenization.
Subsamples (100 μL) of the buffered soil suspension were removed during continuous mixing
using a multi-channel pipette and placed into microplate wells, which were loaded into the
microplate reader. Four replicate wells were filled per substrate. Plates were left to equilibrate
at 30 $^o$C for 5 minutes inside the reader before the automatic addition of 100 μL of substrate,
resulting in a final substrate concentration of 667 μM. Each well was triturated thoroughly
during addition of the substrate. Excitation fluorescence was set at 365 nm. Emission intensity at
450 nm was recorded at 5-minute intervals over a 6-hour period. The effect of fluorescence
quenching was accounted for in each sample by preparing MUF calibration curves in the same
soil suspension as used for the analysis. The limits of detection and quantification were
determined to be 1.1 µM and 3.3 µM MUF respectively, equivalent to 1.1 µM of phosphate for
the determination of phosphomonoesterase activities.

## RESULTS & DISCUSSION

### Sediment characterisation and evidence of bioturbation

Characterisation of sediment cores revealed physical and chemical solid phase homogeneity
within the top 10 cm, with a bulk density of ~1.3 g cm$^{-3}$, water content of ~50% (by weight), OM
of ~3% and a carbonate of ~25% (Figure 2B). Between 10 and 15 cm depth increases in bulk
density and decreases to the sediment water content, OM % and carbonate fraction occurred as
soft sediment transitioned to clay.
A benthic macroinvertebrate density of approximately 49,500 individuals per m$^2$ was
determined, consistent with previously reported values (Pelegri and Blackburn, 1995). The
community (Figure 2A) was dominated by aquatic earthworms (*Tubificidae 60%* and *Branchiura*
*sowerbyii 8%*) which typically feed and mix sediment within the top 5-10 cm (Fisher et al., 1980;
McCall and Fisher, 1980). Other groups identified included *Ceratopogonidae* (No see ums or
biting midges, 22%) including *Sphareomias*, *Probezzia* and *Bezzia, Chironomidae* (Midges, 6%)
including *Cryptochironomus* and *Tanypus, Nemotoda* (round worms, 4%) and a single *Hyalella*
*azteca* (scud <1%).
Bioturbating organisms, such as those identified, have previously been shown to alter
biogeochemical cycling within surface sediments (Hölker et al., 2015). Reported influences
include increased solute fluxes (Furukawa et al., 2001; Matisoff and Wang, 1998), mixing of
solid sediment (Fisher et al., 1980) and bioconveying of sediment particles (Lagauzère et al.,
2009). These processes have been shown to enhance sediment oxygen demand (McCall and
Fisher, 1980; Pelegri and Blackburn, 1995), degradation of OM (Aller, 1994), rates of
denitrification, transport of contaminants to surface water (Lagauzère et al., 2009), and temporal
fluctuation of redox conditions (Aller, 1994). Efficient sediment mixing allows frequent re-
oxidation of reduced sediments and therefore regeneration of terminal electron acceptors (TEAs)
such as nitrate, ferric iron and sulfate, which often limit mineralisation of OM in sediments
underlying hypereutrophic water bodies (Reddy and DeLaune, 2008). Electron donors in the
form of fresh autochthonous necromass are also rapidly redistributed vertically within the zone
of bioturbation. This environment should therefore support a metabolically diverse, abundant and
highly active microbial community (DeAngelis et al., 2010).
Quantitative XRD analysis of the top 12 cm of sediment (Figure 2C) showed close agreement
with the carbonate fraction determined by TGA (~25% by TGA vs 27% by XRD) indicating a
calcite dominated, carbonate buffered system. The remaining mineral assemblage was dominated
by quartz and clay minerals (Illite, 30% and Chamosite, 2%).  No pyrite or vivianite was
detected by XRD suggesting either their absence or presence in low abundance (<1%) with
poorly crystalline structures.
**Experimental redox oscillation: Aqueous chemistry**
$E_h$ within the bioreactor oscillated between +470 and -250 mV (Figure 3) consistent with $E_h$
ranges of wetland sediments (Nikolausz et al., 2008). A slight pH oscillation was also present
between ~7.4 during anoxic half-cycles ($N_2$ sparging) and ~7.7 during oxic-half cycles
($N_2$:$O_2$:$CO_2$ sparging), this variation, shown in Figure 3, is consistent with calcite/dolomite
buffered sediment equilibrating with changing $p_{CO2}$ caused by sparging gas composition and
microbial respiration.  After the 11-day equilibration, ionic strength of the aqueous phase in the
reactor suspension remained at ~0.025 +/- 0.004 M for the duration of the experiment. This range
of $E_h$/pH conditions and ionic strength is consistent with the range measured within surficial
sediment at the field site and transitions across the thermodynamically predicted stability
boundaries for multiple redox couples e.g. $MnO_2/Mn^{2+}$, $NO_3^-/NO_2^-/NH_4^+$, $Fe(OH)_3/Fe^{2+}$, $SO_4^{2-}/HS^-$
, during each 14 day redox cycle. The upper $E_h$ values recorded during oxic cycles are
significantly lower than predicted by the $O_2/H_2O$ couple (820 mV @ pH 7) but are consistent
with the $O_2/H_2O_2$ couple (300 mV @ pH 7) which is considered to control electrode measured $E_h$
under oxic conditions (Stumm and Morgan, 1996).
Aqueous chemistry data, shown in Figure 3, demonstrate the consumption of TEAs in order of
decreasing nominal energetic yield, coupled to the oxidation of labile OM. Upon physical
removal and consumption of residual oxygen by aerobic respiration, nitrate concentration
decreased in the solution. Decreases to nitrate concentration coincided with peaks of nitrite
concentration within the first hour of oxygen removal, indicative of microbial denitrification.
Subsequent increases to $Mn_{(aq)}$, $Fe^{2+}_{(aq)}$ and $HS^-_{(aq)}$ imply sequential reduction of $MnO_2$, $Fe(OH)_3$
and $SO_4^{2-}$ as more energetically efficient electron acceptors were depleted. Mn (predicted as $Mn^{2+}$
by the thermodynamic model) and $Fe^{2+}$ were detected in solution earlier within each subsequent
anoxic cycle, however the apparent order of reduction remained consistent across all five redox
cycles ($O_2$, $NO_3^-$, $NO_2^-$, $MnO_2$, $Fe(OH)_3$, $SO_4^{2-}$). The consistent order and relative magnitude of
reduction implies that the main biogeochemical functioning of the sediment suspension did not
change dramatically between cycles during the experiment.
Although relatively low concentrations of $Fe^{2+}$ (up to 71 $\mu$M) were measured in solution, 0.5 M
HCl extractions revealed that significantly greater $Fe^{2+}$ was produced during each anoxic cycle
than was measured in the aqueous phase. $Fe^{2+}$ generated by dissimilatory iron reduction has been
shown to sorb to mineral surfaces in sediment (Gehin et al., 2007; Klein et al., 2010; Liger et al.,
1999) or precipitate as ferrous carbonate (Jensen et al., 2002), ferrous sulfide or other mixed
ferrous/ferric phases (Rickard and Morse, 2005).  The 0.5M HCl extractions targeted this sorbed
or poorly crystalline freshly precipitated $Fe^{2+}$. During each anoxic cycle HCl extractable $Fe^{2+}$
concentration increased by 50 to 70 $\mu$mol $g^{-1}$, equivalent to 12.31 to 17.29 mM of iron reduction
within the reactor as a whole. Thus, only 0.63 +/- 0.4% of microbially reduced $Fe^{2+}$ was
measureable in solution.  The aqueous phase of the reactor was shown to be supersaturated with
respect to mackinawite (FeS), pyrite ($FeS_2$), vivianite ($Fe_3(PO_4)_2:8H_2O$) and siderite ($FeCO_3$)
during anoxic half-cycles indicating thermodynamic favourability for precipitation of these
minerals.  The kinetic constraints on precipitation were not, however, considered. No significant
cumulative change to extractable $Fe^{2+}/Fe^{3+}$ occurred after five full reduction-oxidation cycles,
indicating that solid phase Fe redox cycling was reversible, potentially due to rapid oxidation of
solid $Fe^{2+}$ in the presence of $O_2$ and carbonate (Caldeira et al., 2010).
DOC concentration also fluctuated systematically during oxic-anoxic cycles (Figure 3). Higher
DOC concentrations were measured during anoxic conditions than oxic conditions.  DOC may
be replenished by both enzymatic hydrolysis of particulate organic matter (POM) (Vetter et al.,
1998) and desorption of mineral associated OM (Grybos et al., 2009). The addition of algal
matter at the beginning of anoxic cycles resulted in observable sharp peaks of DOC which was
rapidly removed from solution, probably due to a combination of mineralization of labile DOC
to $HCO_3^-$ and sorption processes (Chorover and Amistadi, 2001; Grybos et al., 2009).  The peak
of DOC supplied by addition of algal matter represented labile DOC, which was readily
mineralized in comparison to the residual DOC, which persisted in the system throughout the
experiment.   The differences in residual OM mobility between oxic and anoxic cycles were
unlikely due to oxide dissolution as differences to DOC concentration are observed prior to
increases in Mn and Fe concentration in solution. We therefore postulate that solubility changes
to humified DOC were driven by pH changes (Figure 3) between oxic and anoxic conditions
caused by changes in $pCO_2$ between oxic and anoxic conditions as previously shown by Grybos
et al in wetland sediments (Grybos et al., 2009).
Lowest aqueous phosphorus concentrations (~2.5 to 3 $\mu$M), shown in Figure 4, occurred during
oxic half cycles and increased dramatically during anoxic half cycles to a maximum
concentration of 50 to 60 $\mu$M per cycle, 88% of which occurred as SRP. The range of TDP
concentrations within the aqueous phase of the reactor suspension are similar to those reported in
situ at the field site by Mayer et al (2006). The timing of phosphorus release to the aqueous
phase corresponded well with increasing $Fe^{2+}_{(aq)}$ concentration  This is reflected in a strong
positive correlation between TDP and Fe concentrations (n = 37, $R^2$ = 0.93, p < 0.0001).
Increases to aqueous P concentration occurred only after depletion of residual $O_2$, $NO_3^-$ and $NO_2^-$,
after increases to $Mn_{(aq)}$ and before increases to $HS^-_{(aq)}$. The timing of P release is suggestive of an
iron(III)-oxyhydroxide or ferric phosphate control on phosphorus mobility (Bonneville et al.,
2004; Hyacinthe and Van Cappellen, 2004) and indicates that complete nitrate depletion was
required prior to phosphorus release to the aqueous phase.
Full tabulated aqueous chemistry data for the experimental time series is provided as supporting
information.
**Sequential chemical extractions and solid phase P partitioning**
The sum of P concentrations from all sequential extraction reservoirs (61 $\pm$ 5 $\mu$mol g$^{-1}$) was
consistently within 10% of a total P extraction (57 $\pm$ 4 $\mu$mol g$^{-1}$) indicating acceptable analytical
precision from the sequential extraction procedure. Highest P concentrations were associated
with the $P_{Hum}$ (~26% of TP) and $P_{Fe}$ (~24%) fractions with lower P concentrations in the $P_{Ex}$
~16%, $P_{CFA}$ ~15%, $P_{Detr}$ ~12% and $P_{Res}$ ~7% fractions (Figure 4). The largest variations in P
concentration were observed for $P_{Fe}$ and decreased in the order $P_{Fe}$ (2.6) > $P_{Hum}$ (1.5) > $P_{Ex}$ (0.4) >
$P_{Detr}$ (0.4) > $P_{Resi}$ (0.35) > $P_{CFA}$ (0.25), where the numbers in brackets are standard deviations
($\mu$mol g$^{-1}$). High variability over time within the $P_{Fe}$ and $P_{Hum}$ fractions suggests that P was
exchanged to and from these fractions during redox oscillation whereas changes to the $P_{Ex}$, $P_{Resi}$
and $P_{CFA}$ fractions were comparatively insignificant (Figure 4).
The $P_{Fe}$ fraction was the only P pool in which concentration consistently decreased during anoxic
conditions and increased during oxic conditions (Figure 4). When a P mass balance (Figure 5)
was attempted to account for increases to aqueous phosphorus ($P_{Aq}$) from the iron bound ($P_{Fe}$)
pool during anoxic periods it became evident that only approximately 4.5% of variability
observed in the $P_{Fe}$ pool (total $P_{Fe}$ variation of up to 4.5 $\mu$mol g$^{-1}$ during anoxic periods) was
necessary to account for the changes to inter-cycle $P_{Aq}$ concentrations (50 $\mu$M). The remainder of
$P_{Fe}$ lost during anoxic cycles appears to be reversibly redistributed to the $P_{Ex}$ (~30%) and $P_{Hum}$
(~65%) pools which generally increased during anoxic conditions and decreased during oxic
conditions (Figure 4).
According to Ruttenberg (1992) the $P_{Ex}$ fraction corresponds to P mobilised via the formation of
$MgPO_4^-$ complexes and(or) mass action displacement by $Cl^-$. It is therefore considered that
$MgCl_2$ effectively extracts P loosely associated with mineral surfaces. However, Ruttenberg
(1992) also demonstrated that plankton were efficiently extracted by $MgCl_2$ as well as ~25% of P
associated with biogenic $CaCO_3$. Consequently, it is likely that P associated with microbial
biomass, $CaCO_3$, and other loosely sorbed P contributed to the $P_{Ex}$ fraction. Slight increases to $P_{Ex}$
concentration during anoxic periods and corresponding decreases during oxic periods likely
reflect the combination of two processes 1) equilibration between P surface complexes and
aqueous P species due to fluctuating aqueous concentrations which were consistently higher
during anoxic periods (Olsen and Watanabe, 1957) and 2) pH driven sorption/desorption as pH
was consistently slightly higher during aerobic periods (Figure 3) favouring desorption from
mineral surfaces including illite which comprised 30% of the crystalline mineralogical fraction
(Figure 2C) (Manning, 1996).
The $NaHCO_3$ extraction step was originally added to the SEDEX method to target OM
associated P, which would otherwise be liberated during the $P_{Fe}$ extraction step (Baldwin, 1996).
Baldwin noted a brown coloration in $P_{Hum}$ extracts and that absorbance at 250 nm was positively
correlated with SRP. Absorbance at 254 nm has been shown to be indicative of aromatic OM,
commonly associated with humic substances (Weishaar et al., 2003).  A light brown color was
also present in the $NaHCO_3$ extracts recovered during this experiment despite comparatively low
sediment OM content (Figure 2B).  Absorbance spectra for these extracts were not determined.
Li et al (2015) have recently demonstrated that the SEDEX $P_{Fe}$ extraction step co-extracted P
associated with fine iron oxide-OM complexes when a prior $NaHCO_3$ step was not incorporated.
However, Li et al (2015) also suggested that these complexes may be more recalcitrant than pure
minerals. Iron was present within the $P_{Hum}$ extract at concentrations between 18 and 25 $\mu$mol g$^{-1}$
however, the original speciation of this iron is prior to extraction is unknown. Chemically similar
extractions used in soil sciences such as Hedley's extraction (0.5 M $NaHCO_3$, pH 8.5, 16 hours)
and the Olsen-P test (0.5M $NaHCO_3$, pH 8.5, 30 minutes) have been shown to extract Mg and Ca
phosphates as well as some organic P (Hedley et al., 1982; Olsen et al., 1954).  Approximately
2/3 of P extracted within $P_{Hum}$ was present as SRP suggesting that ~1/3 of this fraction may be $P_o$
species. The pH of the $NaHCO_3$ extract used here was adjusted to 7.6 to minimize the dissolution
of Mg and Ca phosphates prior to the $P_{CFA}$ extraction. Despite this, 30 to 44 $\mu$mol g$^{-1}$ of Ca was
extracted within the $P_{Hum}$ fraction. The origin of the extracted Ca is, however, unknown and may
have been complexed with OM or part of labile Ca-phosphates. It is still expected that the
majority of Ca-phosphate minerals were quantified as part of the $P_{CFA}$ or $P_{Detri}$ extractions which
included ~500 to 700 and ~50 to 70 $\mu$mol Ca g$^{-1}$ respectively.
Humic acids are known to compete with orthophosphate for surface binding sites on various
minerals including goethite (Sibanda and Young, 1986) and poorly ordered Fe-oxides in the
short term (Gerke, 1993). However, sorption of natural OM to freshly precipitated Fe oxides may
increase the long term sorption capacity of ferric oxides towards P by decreasing
recrystallization over time (Gerke, 1993) and through the formation of OM-Fe-P complexes
(Gerke, 1993). Although previous studies have provided evidence for ternary complexes between
ferric iron, OM and phosphate (Kizewski et al., 2010b) there is currently no direct spectroscopic
evidence for the existence of mixed OM-Fe(III)-phosphate complexes in natural waters.
Identification of such complexes in natural environments is inherently challenging due to the
complexity of natural geochemical matrices (Kizewski et al., 2010a). Recent studies have
however successfully investigated the structure of synthetic OM-Fe(III)-phosphate complexes
(Kizewski et al., 2010a) and similar OM-Fe(III)-arsenate complexes spectroscopically (Mikutta
and Kretzschmar, 2011; Sharma et al., 2010).  These studies suggest that similar and perhaps
more complex heterogeneous ternary complexes are also likely to be present in natural
freshwater environments (Kizewski et al., 2010a).  This suggestion is also supported by the
observation that more than 80% of soluble P in some natural waters is associated with high
molecular weight OM (Gerke, 2010). As spectroscopic characterization of the P associated with
the $P_{Hum}$ fraction was not performed in this study, the $P_{Hum}$ pool is considered to represent a
variety of OM associated P in addition to small amounts of P from labile Ca-phosphate minerals.
OM associated P extracted within this fraction is likely coordinated with ferric iron (18 to 25
$\mu$mol Fe g$^{-1}$ coextracted), Ca (30 to 44 $\mu$mol Ca g$^{-1}$ coextracted) or Al (0.8 to 1.5 $\mu$mol Al g$^{-1}$
coextracted).  These metals may, in-turn, be associated with various mineral surfaces within the
sediment.
Sequential extraction data demonstrate that the $P_{Hum}$ fraction is the dominant P fraction in all
samples analysed, which highlights the significance of this fraction. P mass balance also suggests
that reversible re-partitioning between this and the $P_{Fe}$ fraction occurs during redox condition
changes.  As the exact chemical nature of the $P_{Hum}$ fraction is not known, interpretation of
concentration changes over time are challenging.  Speculatively, increases to the $P_{Hum}$ fraction
under anoxic conditions may be due to the release of occluded OM-metal-P complexes within
iron(III) oxyhydroxides during reductive dissolution or simply re-equilibration of solid phase
OM-metal-P complexes with increased aqueous P.
We consider that the majority of P extracted within the $P_{Fe}$ fraction was co-precipitated with
iron(III)-oxyhydroxides which were reductively dissolved by dithionite during the extraction
(Ruttenberg, 1992). This interpretation is supported by the relatively high concentrations of iron
extracted within the $P_{Fe}$ fraction (72 to 91 $\mu$mol Fe g$^{-1}$). Aqueous $Fe^{2+}$ produced during this
extraction is subsequently chelated by citrate and therefore solubility of Fe and P is maintained.
The bicarbonate component functions as a pH buffer to ensure maximum preservation of apatite
and $CaCO_3$ bound P during this reaction step (Ruttenberg, 1992).
Neither the $P_{CFA}$, $P_{Detri}$ nor $P_{Resi}$ fractions varied systematically between oxic and anoxic
conditions, or changed consistently during the course of the experiment, suggesting their
comparative stability during short periods of redox fluctuation this is supported by calculated
saturation indices (SI) for hydroxyapatite, which remained between +0.86 and +6.24 for the
duration of the experiment.
The P contribution from individual algal additions ($\sim$1.5 $\mu$mol P g$^{-1}$) was relatively small
compared to total P in the reactor (61 $\mu$mol P g$^{-1}$) and within the margin of analytical error
associated with solid extractions. Additionally, no single fraction shows a clear increase over the
course of the experiment, therefore quantification of the redistribution of P added with algal
additions is not possible.
Full tabulated solid phase chemistry data for the experimental time series is provided as
supporting information.
**Fe:P ratios**
Sequential extraction data, shown in Figure 4, aqueous chemistry data shown in Figure 3, and the
correlation between aqueous Fe and P  (n = 37, R$^2$ = 0.93, p < 0.0001) all suggest that P released
to solution under anoxic conditions originated in the $P_{Fe}$ pool.  Although the maximum molar
ratio for phosphate incorporation within ferric oxides has been shown to be 2:1 (Fe:P) (Thibault
et al., 2009), it has been suggested that much higher solid Fe:P of 15 (Jensen et al., 1992) to >20
(Phillips et al., 1994) maybe necessary to control phosphorus mobility under oxic conditions.
Results from bioreactor experiments suggest that phosphorus is retained in the solid phase under
oxic conditions at total Fe:P ratios of just 4.1:1, potentially due to the association of P with other
solid sedimentary pools, particularly $P_{Hum}$. Fe:P ratios below the stoichiometric limitation of 2:1,
measured in the aqueous phase (1.5 to 1.9), during anoxic conditions are therefore likely due to
the removal of Fe$^{2+}$ from solution by secondary sorption and precipitation processes, subsequent
to reductive dissolution.  Probable secondary Fe$^{2+}$ removal processes include the formation of
amorphous FeS (SI of up to +2.27 for mackinawite) (Rickard and Morse, 2005) and sorption of
Fe$^{2+}$ to clays (Gehin et al., 2007; Klein et al., 2010). This is supported by increases to Fe$^{2+}$/Fe$^{3+}$
ratios in the 0.5 M HCl extractable fraction during anoxic conditions. Frequent rapid reoxidation
of ferrous sulfide due to air sparging in the reactor experiment, and extensive bioturbation in-
situ, likely prevents formation of more recalcitrant and slow forming iron sulfide minerals such
as pyrite, despite strong thermodynamic favourability for pyrite formation (SI up to +13.13)
(Caldeira et al., 2010; Peiffer et al., 2015). This is consistent with the results of XRD analysis,
which did not identify pyrite (Figure 2C). A similar effect has been previously demonstrated in
lake sediments (Gächter and Müller, 2003).

## Hydrolytic enzyme activities

The activities of model phosphomonoesterases, phosphodiesterases and pyrophosphatase were
found to be systematically higher under oxic conditions compared to anoxic conditions by 37%
(p < 0.005), 8% (not significant) and 24% (p = 0.08) respectively (Figure 6B).
Phosphomonoesterases were found to have the highest activities despite the inherent
overestimation of phosphodiesterase activities when using MUF tagged substrates (Sirová et al.,
2013). The opposite trend was observed for glycopyranoside, part of the cellulose degradation
pathway (Dunn et al., 2013), which showed consistently higher activity (69% p = <0.05) under
anoxic conditions (Figure 6B). The different trends exhibited by cellulose and phosphatase
enzymes, indicate that changes in activity were not universal but specific to enzyme function.
Phosphomonoesterase activities obtained in the current study (1.76 to 2.4 mmol h$^{-1}$ kg$^{-1}$) are
similar to those previously reported in wetland sediments (Kang and Freeman, 1999) and suggest
the capacity for rapid hydrolysis of P$_o$ species in necromass. Lowering of the water table in
wetlands has previously been shown to increase the activity of phosphatase enzymes and the
hydrolysis of P$_o$ species (Song et al., 2007). However, water table fluctuation results in
concomitant changes to moisture content and redox conditions which prevents isolation of the
causal variable in field investigations (Rezanezhad et al., 2014). Therefore, this is to the best of
our knowledge, the first direct demonstration of phosphatase activity changes in response to
changing redox conditions. We postulate that under anoxic conditions when phosphorus
availability in the aqueous phase is high, production of extracellular phosphatase enzymes by the
microbial community is down regulated. Conversely, when bioavailable phosphorus is removed
from solution under oxic conditions, extracellular phosphatase production is up regulated in
response. Adjustments to enzyme production in response to changes in phosphate availability
must occur on short timescales (hours/days) for such trends to be observable during the
bioreactor experiment. An inverse relationship between phosphatase activities and phosphate
concentration, has previously been shown spatially in wetlands by Kang and Freeman (1999) but
to our knowledge never temporally in sediments.
**$^{31}$P NMR**
Results from $^{31}$P NMR analyses (Figure 6A) show that the majority of phosphorus was present in
the solid phase as ortho-phosphate (84-91%) with 4-8% monoester P, 3-8% diester P and <1%
phosphonates and polyphosphates with no clear trend in relative abundance emerging during the
experiment. The NaOH-EDTA extraction resulted in a recovery of ~27% of TP, which is
comparable to previous studies with carbonate buffered soils and sediments (Hansen et al., 2004;
Turner et al., 2003a). Alpha and beta glycerophosphates are commonly identified in monoester
spectral regions and have been demonstrated to be products of diesters degraded during analysis
(Doolette et al., 2009; Jørgensen et al., 2015; Paraskova et al., 2015). As no glycerophosphates
were identified in any of the analysed samples, recalculation of monoester/diester ratios was not
performed. A higher mean monoester/diester ratio (2.31) was found in reduced samples than
oxidised samples (0.97) a statistically significant difference (p=0.04). This difference could
indicate that monoester P was either less efficiently extracted under oxic conditions due to
sorption to metal oxides or that monoesterase/diesterase activity decreased under anoxic
conditions, which is consistent with enzymatic activity assays (Figure 6B).    Total $P_o$ determined
by $^{31}$P NMR varied between 9 and 16% over time compared to 5 to 11% in the $P_{resi}$ from
sequential extractions, indicating that not all $P_o$ was extracted in the $P_{resi}$ fraction, which is
commonly referred to as the organic-P fraction. We postulate that the remaining ~5% of total
phosphorus, identified as $P_o$ by $^{31}$P NMR was extracted during previous steps in the sequential
extraction scheme, particularly $MgCl_2$, which has been shown to efficiently extract P associated
with microbial biomass (Ruttenberg, 1992) and $NaHCO_3$. The relative activities of phosphatase
enzymes appear to correlate with the relative abundances of $P_o$ species identified by $^{31}$P NMR
e.g. Monoesters > Diesters > Pyro-P.
Significant polyphosphate (> 1%) was not detected by $^{31}$P NMR during experiments.  Previous
studies focusing on WWTP tertiary treatment for phosphate removal suggest that redox
oscillating conditions promote intracellular poly-P accumulation during aerobic conditions to be
used as an energy store under anoxic conditions in order to uptake short chain fatty acids (SCFA)
in the absence of an electron acceptor (Hupfer et al., 2007; Wentzel et al., 1991). Phosphate
uptake during aerobic conditions therefore requires P availability in excess of what is required
for growth and maintenance of the microbial community.  However, phosphate availability under
aerobic conditions is limited by sorption and coprecipitation with iron oxides, assuming
sufficient Fe:P, despite high total phosphorus concentration in sediment.  The P requirements by
the microbial community are also likely to be high during the transition to aerobic conditions due
to the availability of $O_2$ as an energetically efficient electron acceptor and fermentation products
(SCFA), further decreasing the probability of polyphosphate accumulation. Additionally,
polyphosphate accumulation and release has shown to be inhibited by denitrification and sulfate
reduction due to competition for SCFA (Kortstee et al., 1994; Yamamoto-Ikemoto et al., 1994).
**IMPLICATIONS**
Our controlled laboratory simulation of highly dynamic redox conditions in eutrophic sediment
demonstrates the importance of multiple coupled elemental cycles (C, N, Fe, S, P) when
determining internal P loading potential and timing. Our results demonstrate that neither aqueous
or solid phase Fe:P ratios or even solid phase $P_{Fe}$ quantification are good predictors of potential P
release to the water column under anoxic conditions, due to extensive reversible redistribution of
both reduced Fe and associated P within the solid phase. We show that 99.4% of reduced Fe and
95.5% of $P_{Fe}$ are not released to the aqueous phase upon Fe reduction but reversibly redistributed
within the solid phase upon short periods of iron reduction. Additionally, the apparent
requirement for complete nitrate depletion prior to anoxia-promoted P release to the aqueous
phase has potential implications for water bodies where significant iron bound legacy P is
present within sediments. Our results suggest that decreasing $NO_3^-$ concentrations in external
loads, while ostensibly ecologically beneficial, may, in some cases, increase the frequency and
magnitude of internal P loading during short periods of anoxia. In P limited systems, the
apparent ecological benefits of decreased $NO_3^-$ may be offset by increased P release and
eutrophication. However, numerous additional processes exist in natural systems, which were
not simulated during our reactor experiment and may influence internal loading mechanisms.
Finally, we demonstrate that oscillatory redox conditions, even in sediments with diverse and
active microbial communities, do not necessarily result in accumulation of polyphosphate, due to
mineralogical phosphate immobilization and scavenging of SCFA by anaerobic heterotrophic
respiration.

## Funding Sources

We acknowledge funding from the Canadian Excellence Research Chair (CERC) program and
the Water Institute at the University of Waterloo.

## Acknowledgements

We would like to acknowledge the support of the Royal Botanical Gardens, particularly Jennifer
Bowman and Tys Theymeyer as well as Taylor Maavara who kindly provided her pack raft for
use during field sampling.  We also acknowledge Jia Cheng (Allen) Yu for his assistance with
the production of Figure 1 as well as Marianne Vandergriendt, Kassandra Ma and Christine
Ridenour for laboratory assistance.

## Abbreviations

Green filamentous algae; GFA, waste water treatment plants; WWTP, sediment-water interface;
SWI, dissolved organic carbon; DOC, organic phosphorus species; $P_o$, soluble reactive
phosphorus; SRP, Inductively Coupled Plasma Optical Emission Spectrometry; ICP-OES,
organic matter; OM, powder X-ray diffraction; XRD, dissolved reactive phosphorus; DRP,
Method detection limit; MDL, non-purgeable organic carbon; NPOC, total dissolved
phosphorus; TDP, dissolved inorganic carbon; DIC, relative standard deviation; RSD%, 4-
methylumbelliferyl; MUF, 4-Methylumbelliferyl phosphate; MUP, Bis(4-
methylumbelliferyl)phosphate; DiMUP, and 4-Methylumbelliferyl pyrophosphate; PYRO-P, 4-
Methylumbelliferyl beta-D-glucopyranoside; MUGb, thermo-gravimetric analysis; TGA,
terminal electron acceptors; TEAs, particulate organic matter; POM, total phosphorus; TP.

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

Methods, Agronomy. Soil Science Society of America, Madison, Wisconsin, pp. 493–
544.
Gehin, A., Greneche, J.-M., Tournassat, C., Brendle, J., Rancourt, D.G., Charlet, L., 2007.
Reversible surface-sorption-induced electron-transfer oxidation of Fe(II) at reactive sites
on a synthetic clay mineral. Geochim. Cosmochim. Acta 71, 863–876. doi:doi:
10.1016/j.gca.2006.10.019
Gerke, J., 2010. Humic (Organic Matter)-Al(Fe)-Phosphate Complexes: An Underestimated
Phosphate Form in Soils and Source of Plant-Available Phosphate. Soil Sci. 175, 417–
425. doi:10.1097/SS.0b013e3181f1b4dd
Gerke, J., 1993. Phosphate adsorption by humic/Fe-oxide mixtures aged at pH 4 and 7 and by
poorly ordered Fe-oxide. Geoderma 59, 279–288. doi:10.1016/0016-7061(93)90074-U
Gorham, E., Boyce, F.M., 1989. Influence of Lake Surface Area and Depth Upon Thermal
Stratification and the Depth of the Summer Thermocline. J. Gt. Lakes Res. 15, 233–245.
doi:10.1016/S0380-1330(89)71479-9
Grybos, M., Davranche, M., Gruau, G., Petitjean, P., Pedrot, M., 2009. Increasing pH drives
organic matter solubilization from wetland soils under reducing conditions. Geoderma
154, 13–19. doi:doi: DOI: 10.1016/j.geoderma.2009.09.001
Hansen, J.C., Cade-Menun, B.J., Strawn, D.G., 2004. Phosphorus Speciation in Manure-
Amended Alkaline Soils. J. Environ. Qual. 33, 1521. doi:10.2134/jeq2004.1521
Hedley, M.J., White, R.E., Nye, P.H., 1982. Plant-Induced changes in the rhizosphere of rape
(Brassica Napus Var.Emerald) seedlings. III. Changes in L value, soil phosphate fractions
and phosphatase activity. New Phytol. 91, 45–56. doi:10.1111/j.1469-
8137.1982.tb03291.x
Hölker, F., Vanni, M.J., Kuiper, J.J., Meile, C., Grossart, H.-P., Stief, P., Adrian, R., Lorke, A.,
Dellwig, O., Brand, A., Hupfer, M., Mooij, W.M., Nützmann, G., Lewandowski, J., 2015.
Tube-dwelling invertebrates: tiny ecosystem engineers have large effects in lake
ecosystems. Ecol. Monogr. 85, 333–351. doi:10.1890/14-1160.1
Hongve, D., 1997. Cycling of iron, manganese, and phosphate in a meromictic lake. Limnol.
Oceanogr. 42, 635–647. doi:10.4319/lo.1997.42.4.0635
Hupfer, M., Gloess, S., Grossart, H., 2007. Polyphosphate-accumulating microorganisms in
aquatic sediments. Aquat. Microb. Ecol. 47, 299–311. doi:10.3354/ame047299
Hyacinthe, C., Van Cappellen, P., 2004. An authigenic iron phosphate phase in estuarine
sediments: composition, formation and chemical reactivity. Mar. Chem. 91, 227–251.
doi:10.1016/j.marchem.2004.04.006
Jensen, D.L., Boddum, J.K., Tjell, J.C., Christensen, T.H., 2002. The solubility of rhodochrosite
(MnCO3) and siderite (FeCO3) in anaerobic aquatic environments. Appl. Geochem. 17,
503–511. doi:doi: DOI: 10.1016/S0883-2927(01)00118-4
Jensen, H.S., Kristensen, P., Jeppesen, E., Skytthe, A., 1992. Iron:phosphorus ratio in surface
sediment as an indicator of phosphate release from aerobic sediments in shallow lakes.
Hydrobiologia 235–236, 731–743. doi:10.1007/BF00026261
Jørgensen, C., Inglett, K.S., Jensen, H.S., Reitzel, K., Reddy, K.R., 2015. Characterization of
biogenic phosphorus in outflow water from constructed wetlands. Geoderma 257–258,
58–66. doi:10.1016/j.geoderma.2015.01.019
Joshi, S.R., Kukkadapu, R.K., Burdige, D.J., Bowden, M.E., Sparks, D.L., Jaisi, D.P., 2015.
Organic Matter Remineralization Predominates Phosphorus Cycling in the Mid-Bay

Sediments in the Chesapeake Bay. Environ. Sci. Technol. 49, 5887–5896. doi:10.1021/es5059617

Kang, H., Freeman, C., 1999. Phosphatase and arylsulphatase activities in wetland soils: annual variation and controlling factors. Soil Biol. Biochem. 31, 449–454. doi:10.1016/S0038-0717(98)00150-3

Katsev, S., Tsandev, I., L'Heureux, I., Rancourt, D.G., 2006. Factors controlling long-term phosphorus efflux from lake sediments: Exploratory reactive-transport modeling. Chem. Geol. 234, 127–147. doi:10.1016/j.chemgeo.2006.05.001

Kim, D.-K., Zhang, W., Rao, Y.R., Watson, S., Mugalingam, S., Labencki, T., Dittrich, M., Morley, A., Arhonditsis, G.B., 2013. Improving the representation of internal nutrient recycling with phosphorus mass balance models: A case study in the Bay of Quinte, Ontario, Canada. Ecol. Model. 256, 53–68. doi:10.1016/j.ecolmodel.2013.02.017

Kizewski, F.R., Boyle, P., Hesterberg, D., Martin, J.D., 2010a. Mixed Anion (Phosphate/Oxalate) Bonding to Iron(III) Materials. J. Am. Chem. Soc. 132, 2301–2308. doi:10.1021/ja908807b

Kizewski, F.R., Hesterberg, D., Martin, J., 2010b. Phosphate sorption to organic matter/ferrihydrite systems as affected by aging time. Presented at the 19th World Congress of Soil Science, Soil Solutions for a Changing World, Brisbane, Australia.

Klein, A.R., Baldwin, D.S., Singh, B., Silvester, E.J., 2010. Salinity-induced acidification in a wetland sediment through the displacement of clay-bound iron(II). Environ. Chem. 7, 413. doi:10.1071/EN10057

Kortstee, G.J.J., Appeldoorn, K.J., Bonting, C.F.C., Niel, E.W.J., Veen, H.W., 1994. Biology of polyphosphate-accumulating bacteria involved in enhanced biological phosphorus removal. FEMS Microbiol. Rev. 15, 137–153. doi:10.1111/j.1574-6976.1994.tb00131.x

Lagauzère, S., Boyer, P., Stora, G., Bonzom, J.-M., 2009. Effects of uranium-contaminated sediments on the bioturbation activity of Chironomus riparius larvae (Insecta, Diptera) and Tubifex tubifex worms (Annelida, Tubificidae). Chemosphere 76, 324–334. doi:10.1016/j.chemosphere.2009.03.062

Li, W., Joshi, S.R., Hou, G., Burdige, D.J., Sparks, D.L., Jaisi, D.P., 2015. Characterizing Phosphorus Speciation of Chesapeake Bay Sediments Using Chemical Extraction, [31] P NMR, and X-ray Absorption Fine Structure Spectroscopy. Environ. Sci. Technol. 49, 203–211. doi:10.1021/es504648d

Liger, E., Charlet, L., Van Cappellen, P., 1999. Surface catalysis of uranium(VI) reduction by iron(II). Geochim. Cosmochim. Acta 63, 2939–2955. doi:doi: DOI: 10.1016/S0016-7037(99)00265-3

Mallin, M.A., McIver, M.R., Wells, H.A., Parsons, D.C., Johnson, V.L., 2005. Reversal of eutrophication following sewage treatment upgrades in the New River Estuary, North Carolina. Estuaries 28, 750–760. doi:10.1007/BF02732912

Manning, B.A., 1996. Modeling Arsenate Competitive Adsorption on Kaolinite, Montmorillonite and Illite. Clays Clay Miner. 44, 609–623. doi:10.1346/CCMN.1996.0440504

Matisoff, G., Kaltenberg, E.M., Steely, R.L., Hummel, S.K., Seo, J., Gibbons, K.J., Bridgeman, T.B., Seo, Y., Behbahani, M., James, W.F., Johnson, L.T., Doan, P., Dittrich, M., Evans, M.A., Chaffin, J.D., 2016. Internal loading of phosphorus in western Lake Erie. J. Gt. Lakes Res. 42, 775–788. doi:10.1016/j.jglr.2016.04.004

Matisoff, G., Wang, X., 1998. Solute transport in sediments by freshwater infaunal bioirrigators. Limnol. Oceanogr. 43, 1487–1499. doi:10.4319/lo.1998.43.7.1487

Mayer, T., Rosa, F., Mayer, R., Charlton, M., 2006. Relationship Between the Sediment
Geochemistry and Phosphorus Fluxes in a Great Lakes Coastal Marsh, Cootes Paradise,
ON, Canada. Water Air Soil Pollut. Focus 6, 495–503. doi:10.1007/s11267-006-9033-6

McCall, P.L., Fisher, J.B., 1980. Effects of Tubificid Oligochaetes on Physical and Chemical
Properties of Lake Erie Sediments, in: Brinkhurst, R., Cook, D. (Eds.), Aquatic
Oligochaete Biology. Springer US, pp. 253–317.

McLaughlin, C., Pike, K., 2014. Muddied Waters: The Ongoing Challenge of Sediment and
Phosphorus for Hamilton Harbour Remediation (2014 Towards Safe Harbour Report).
Bay Area Restoration Council, Hamilton, Ontario.

Mikutta, C., Kretzschmar, R., 2011. Spectroscopic Evidence for Ternary Complex Formation
between Arsenate and Ferric Iron Complexes of Humic Substances. Environ. Sci.
Technol. 45, 9550–9557. doi:10.1021/es202300w

Mortimer, C.H., 1971. Chemical Exchanges Between Sediments and Water in the Great Lakes-
Speculations on Probable Regulatory Mechanisms. Limnol. Oceanogr. 16, 387–404.
doi:10.2307/2834170

Mortimer, C.H., 1941. The exchange of dissolved substances between mud and water in lakes. J.
Ecol. 29, 280–329.

Murphy, J., Riley, J.P., 1962. A modified single solution method for the determination of
phosphate in natural waters. Anal. Chim. Acta 27, 31–36. doi:10.1016/S0003-
2670(00)88444-5

Nikolausz, M., Kappelmeyer, U., Szekely, A., Rusznyak, A., Marialigeti, K., Kastner, M., 2008.
Diurnal redox fluctuation and microbial activity in the rhizosphere of wetland plants. Eur.
822           J. Soil Biol. 44, 324–333. doi:doi: DOI: 10.1016/j.ejsobi.2008.01.003

Nürnberg, G.K., 1988. Prediction of Phosphorus Release Rates from Total and Reductant-
Soluble Phosphorus in Anoxic Lake Sediments. Can. J. Fish. Aquat. Sci. 45, 453–462.
doi:10.1139/f88-054

O'Connell, D.W., Jensen, M.M., Jakobsen, R., Thamdrup, B., Andersen, T.J., Kovacs, A.,
Hansen, H.C.B., 2015. Vivianite formation and its role in phosphorus retention in Lake
Ørn, Denmark. Chem. Geol. doi:10.1016/j.chemgeo.2015.05.002

Olsen, S.R., Cole, C.V., Watanabe, F.S., Dean, L.A., 1954. Estimation of available phosphorus
in soils by extraction with sodium bicarbonate. US Dep Agric Circ 939, 1–19.

Olsen, S.R., Watanabe, F.S., 1957. A Method to Determine a Phosphorus Adsorption Maximum
of Soils as Measured by the Langmuir Isotherm1. Soil Sci. Soc. Am. J. 21, 144.
doi:10.2136/sssaj1957.03615995002100020004x

Painter, D.., Hampton, L., Simser, W.L., 1991. Cootes Paradise Water Turbidity: Sources and
Recommendations., in: NWRI Contribution Paper #91-15. Burlington, Ontario, p. 18.

Pallasser, R., Minasny, B., McBratney, A.B., 2013. Soil carbon determination by
thermogravimetrics. PeerJ 1, e6. doi:10.7717/peerj.6

Paraskova, J.V., Jørgensen, C., Reitzel, K., Pettersson, J., Rydin, E., Sjöberg, P.J.R., 2015.
Speciation of Inositol Phosphates in Lake Sediments by Ion-Exchange Chromatography
Coupled with Mass Spectrometry, Inductively Coupled Plasma Atomic Emission
Spectroscopy, and [31] P NMR Spectroscopy. Anal. Chem. 87, 2672–2677.
doi:10.1021/ac5033484

Parkhurst, D.L., Appelo, C.A.J., (US), G.S., 1999. User's guide to PHREEQC (Version 3): A
computer program for speciation, batch-reaction, one-dimensional transport, and inverse
geochemical calculations. US Geological Survey Reston, VA.

Parsons, C.T., Couture, R.-M., Omoregie, E.O., Bardelli, F., Greneche, J.-M., Roman-Ross, G.,
Charlet, L., 2013. The impact of oscillating redox conditions: Arsenic immobilisation in
contaminated calcareous floodplain soils. Environ. Pollut. 178, 254–263.
doi:10.1016/j.envpol.2013.02.028
Peiffer, S., Behrends, T., Hellige, K., Larese-Casanova, P., Wan, M., Pollok, K., 2015. Pyrite
formation and mineral transformation pathways upon sulfidation of ferric hydroxides
depend on mineral type and sulfide concentration. Chem. Geol. 400, 44–55.
doi:10.1016/j.chemgeo.2015.01.023
Pelegri, S.P., Blackburn, T.H., 1995. Effects of Tubifex tubifex (Oligochaeta: Tubificidae) on N-
mineralization in freshwater sediments, measured with isotopes. Aquat. Microb. Ecol. 9,
289–294.
Penn, M.R., Auer, M.T., Doerr, S.M., Driscoll, C.T., Brooks, C.M., Effler, S.W., 2000.
Seasonality in phosphorus release rates from the sediments of a hypereutrophic lake
under a matrix of pH and redox conditions. Can. J. Fish. Aquat. Sci. 57, 1033–1041.
doi:10.1139/f00-035
Phillips, G., Jackson, R., Bennett, C., Chilvers, A., 1994. The importance of sediment
phosphorus release in the restoration of very shallow lakes (The Norfolk Broads,
England) and implications for biomanipulation. Hydrobiologia 275–276, 445–456.
doi:10.1007/BF00026733
Pomeroy, R., 1954. Auxiliary Pretreatment by Zinc Acetate in Sulfide Analyses. Anal. Chem.
26, 571–572. doi:10.1021/ac60087a047
Pretty, J.N., Mason, C.F., Nedwell, D.B., Hine, R.E., Leaf, S., Dils, R., 2003. Environmental
Costs of Freshwater Eutrophication in England and Wales. Environ. Sci. Technol. 37,
201–208. doi:10.1021/es020793k
Reddy, K.R., DeLaune, R.D., 2008. Biogeochemistry of wetlands: science and applications.
CRC Press, Boca Raton.
Reddy, K.R., Newman, S., Osborne, T.Z., White, J.R., Fitz, H.C., 2011. Phosphorous Cycling in
the Greater Everglades Ecosystem: Legacy Phosphorous Implications for Management
and Restoration. Crit. Rev. Environ. Sci. Technol. 41, 149–186.
doi:10.1080/10643389.2010.530932
Reitzel, K., Ahlgren, J., DeBrabandere, H., Waldebäck, M., Gogoll, A., Tranvik, L., Rydin, E.,
2007. Degradation rates of organic phosphorus in lake sediment. Biogeochemistry 82,
15–28. doi:10.1007/s10533-006-9049-z
Rezanezhad, F., Couture, R.-M., Kovac, R., O'Connell, D., Van Cappellen, P., 2014. Water table
fluctuations and soil biogeochemistry: An experimental approach using an automated soil
column system. J. Hydrol. 509, 245–256. doi:10.1016/j.jhydrol.2013.11.036
Rickard, D., Morse, J.W., 2005. Acid volatile sulfide (AVS). Mar. Chem. 97, 141–197.
doi:10.1016/j.marchem.2005.08.004
Routledge, I., 2012. City of Hamilton: King Street (Dundas) Wastewater Treatment Plant. 2011
Annual Report (Annual Report No. Works Number 120001372). The City of Hamilton,
Environment and Sustainable Infrastructure Division, Hamilton, Ontario.
Ruttenberg, K.C., 1992. Development of a sequential extraction method for different forms of
phosphorus in marine sediments. Limnol. Oceanogr. 37, 1460–1482.
doi:10.4319/lo.1992.37.7.1460
Sannigrahi, P., Ingall, E., 2005. Polyphosphates as a source of enhanced P fluxes in marine
sediments overlain by anoxic waters: Evidence from [sup 31]P NMR. Geochem. Trans. 6,
52. doi:10.1063/1.1946447

Semkin, R.G., McLarty, A.W., Craig, D., 1976. A water quality study of Cootes Paradise.
Ontario Ministry of Environment, West Central Region, Toronto, Ontario.

Sharma, P., Ofner, J., Kappler, A., 2010. Formation of Binary and Ternary Colloids and
Dissolved Complexes of Organic Matter, Fe and As. Environ. Sci. Technol. 44, 4479–
4485. doi:10.1021/es100066s

Sharpley, A.N., Chapra, S.C., Wedepohl, R., Sims, J.T., Daniel, T.C., Reddy, K.R., 1994.
Managing Agricultural Phosphorus for Protection of Surface Waters: Issues and Options.
900       J. Environ. Qual. 23, 437. doi:10.2134/jeq1994.00472425002300030006x

Sibanda, H.M., Young, S.D., 1986. Competitive adsorption of humus acids and phosphate on
goethite, gibbsite and two tropical soils. J. Soil Sci. 37, 197–204. doi:10.1111/j.1365-
2389.1986.tb00020.x

Sirová, D., Rejmánková, E., Carlson, E., Vrba, J., 2013. Current standard assays using artificial
substrates overestimate phosphodiesterase activity. Soil Biol. Biochem. 56, 75–79.
doi:10.1016/j.soilbio.2012.02.008

Smith, V.H., Schindler, D.W., 2009. Eutrophication science: where do we go from here? Trends
Ecol. Evol. 24, 201–207. doi:10.1016/j.tree.2008.11.009

Søndergaard, M., Jensen, J.P., Jeppesen, E., 2003. Role of sediment and internal loading of
phosphorus in shallow lakes. Hydrobiologia 506–509, 135–145.
doi:10.1023/B:HYDR.0000008611.12704.dd

Song, K.-Y., Zoh, K.-D., Kang, H., 2007. Release of phosphate in a wetland by changes in
hydrological regime. Sci. Total Environ. 380, 13–18. doi:10.1016/j.scitotenv.2006.11.035

Stookey, L.L., 1970. Ferrozine - a new spectrophotometric reagent for iron. Anal. Chem. 42,
779–781. doi:doi: 10.1021/ac60289a016

Stumm, W., Morgan, J.J., 1996. Aquatic Chemistry, 3rd ed. ed. John Wiley & Sons, New York
[etc.].

Theysmeyer, T., Smith, T., Simser, L., 1999. West Pond 1999 Study. Royal Botanical Gardens,
Science Department.

Thibault, P.-J., Rancourt, D.G., Evans, R.J., Dutrizac, J.E., 2009. Mineralogical confirmation of
a near-P:Fe=1:2 limiting stoichiometric ratio in colloidal P-bearing ferrihydrite-like
hydrous ferric oxide. Geochim. Cosmochim. Acta 73, 364–376.
doi:10.1016/j.gca.2008.10.031

Thompson, A., Chadwick, O.A., Rancourt, D.G., Chorover, J., 2006. Iron-oxide crystallinity
increases during soil redox oscillations. Geochim. Cosmochim. Acta 70, 1710–1727.
doi:10.1016/j.gca.2005.12.005

Turner, B.L., Cade-Menun, B.J., Westermann, D.T., 2003a. Organic phosphorus composition
and potential bioavailability in semi-arid arable soils of the Western United States. Soil
Sci. Soc. Am. J. 67, 1168–1179.

Turner, B.L., Mahieu, N., Condron, L., 2003b. Phosphorus-31 Nuclear Magnetic Resonance
Spectral Assignments of Phosphorus Compounds in Soil NaOH–EDTA Extracts. Soil
Sci. Soc. Am. J. 67, 497. doi:10.2136/sssaj2003.4970

U.S. EPA, 9/99. Field Sampling Guidance Document: Sediment Sampling (No. #1215). U.S.
Environmental Protection Agency. Region 9 Laboratory, Richmond California.

Vetter, Y.A., Deming, J.W., Jumars, P.A., Krieger-Brockett, B.B., 1998. A Predictive Model of
Bacterial Foraging by Means of Freely Released Extracellular Enzymes. Microb. Ecol.
36, 75–92. doi:10.1007/s002489900095
Viollier, E., Inglett, P.., Hunter, K., Roychoudhury, A.., Van Cappellen, P., 2000. The ferrozine
method revisited: Fe(II)/Fe(III) determination in natural waters. Appl. Geochem. 15,
785–790. doi:10.1016/S0883-2927(99)00097-9
Weishaar, J.L., Aiken, G.R., Bergamaschi, B.A., Fram, M.S., Fujii, R., Mopper, K., 2003.
Evaluation of Specific Ultraviolet Absorbance as an Indicator of the Chemical
Composition and Reactivity of Dissolved Organic Carbon. Environ. Sci. Technol. 37,
4702–4708. doi:10.1021/es030360x
Wentzel, M.C., Lötter, L.H., Ekama, G.A., Loewenthal, R.E., Marais, G. v R., 1991. Evaluation
of Biochemical Models for Biological Excess Phosphorus Removal. Water Sci. Technol.
23, 567–576.
Yamamoto-Ikemoto, R., Matsui, S., Komori, T., 1994. Ecological interactions among
denitrification, poly-P accumulation, sulfate reduction, and filamentous sulfur bacteria in
activated sludge. Water Sci. Technol. 30, 201–210.

**Tables**

| Step | Extractant | Conditions | Target phase |
|---|---|---|---|
| 1a | 1M $MgCl_2$ | pH 8 for 2 hours @ 25°C | Exchangeable or Loosely sorbed P ($P_{Ex}$) |
| 1b | 1M $MgCl_2$ | pH 8 for 2 hours @ 25°C | |
| 1c | 18.2 MΩ cm$^{-1}$ $H_2O$ | 2 hours @ 25°C | |
| 2a | 1M $NaHCO_3$ | pH 7.6 for 16 hours @ 25°C | Organic associated P ($P_{Hum}$) |
| 2b | 1M $NaHCO_3$ | pH 7.6 for 2 hours @ 25°C | |
| 2c | 1M $NaHCO_3$ | pH 7.6 for 2 hours @ 25°C | |
| 2d | 1M $NaHCO_3$ | pH 7.6 for 2 hours @ 25°C | |
| 2e | 1M $MgCl_2$ | pH 8 for 2 hours @ 25°C | |
| 3a | 0.3 M $Na_3$-citrate with 1M $NaHCO_3$ with 1.125g of Na-ditionite (CDB) | pH 7.6 for 8 hours @ 25°C | Fe-bound P ($P_{Fe}$) |
| 3b | CDB | pH 7.6 for 2 hours @ 25°C | |
| 3c | 1M $MgCl_2$ | pH 8 for 2 hours @ 25°C | |
| 4a | 1M Na-acetate with acetic acid | pH 4 for 6 hours @ 25°C | Authigenic carbonate fluorapatite plus biogenic apatite plus $CaCO_3$-bound P ($P_{CFA}$) |
| 4b | 1M Na-acetate with acetic acid | pH 4 for 2 hours @ 25°C | |
| 4c | 1M $MgCl_2$ | pH 8 for 2 hours @ 25°C | |
| 5 | 1M HCl | 16 hours @ 25°C | Detrital Apatite plus other inorganic-P ($P_{Detri}$) |

| 6 | 1M HCl | 16 hours after ashing at 550ºC @ 25ºC | Residual/Organic-P ($P_{Resi}$) |
|---|---|---|---|


*Table 1: Summary of the modified SEDEX sequential extraction protocol used on solid samples taken over a time series during the reactor experiment.  Results of this extraction are shown in Figure 4.*

**Figures**

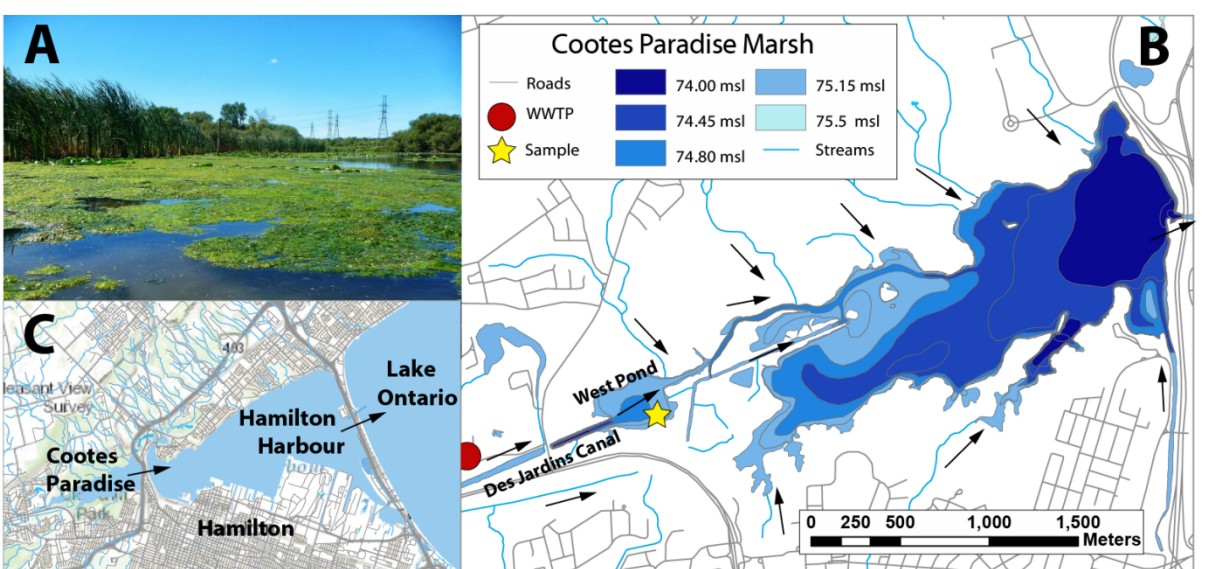


*Figure 1: A) Photograph of the sampling location taken on the day of sampling illustrating the abundance of green filamentous algae. B) Map of Cootes Paradise and West Pond showing the sampling location, local hydrological network and the King Street Waste Water Treatment plant in Dundas. Color represents area covered by surface water at different water levels (msl= meters above sea level) C) Overview map showing the hydrological connection between Cootes Paradise, Hamilton Harbour and Lake Ontario.*

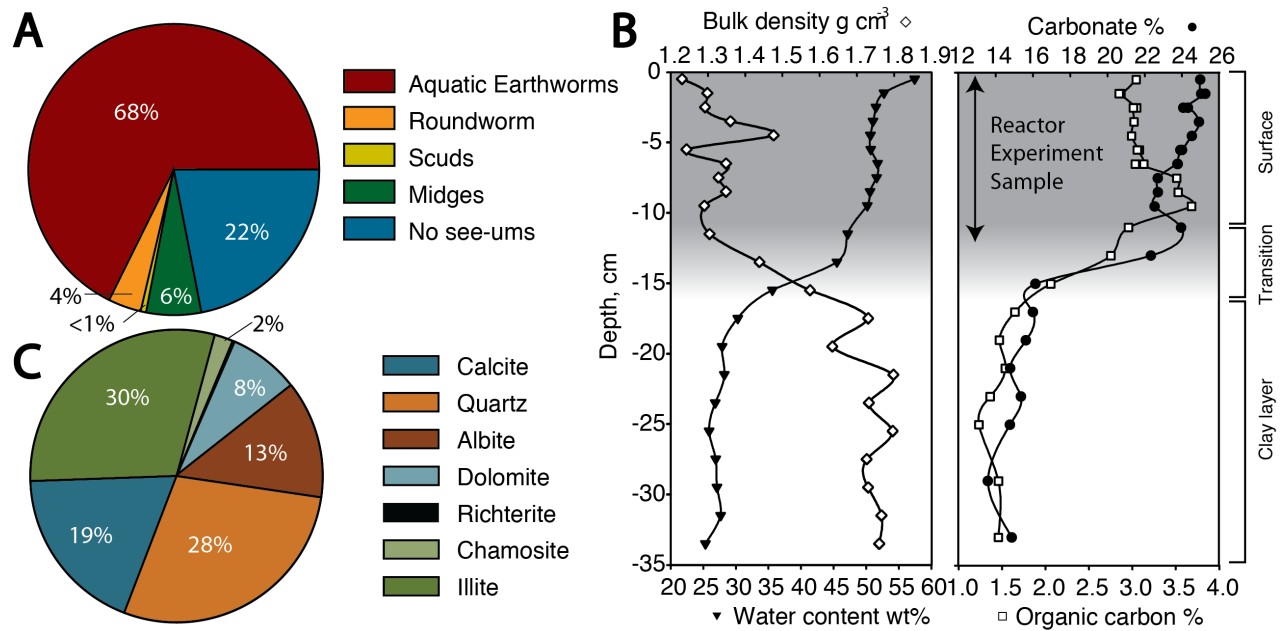

*Figure 2: a) Proportions of bioturbating macro invertebrates identified in the top*
*18 cm b) Depth profiles of sampled sediment, water content weight % (inverted*
*black triangles), bulk density (white diamonds) OM % (white squares), carbonate*
*% (black circles) c) Mineralogical composition of sediments from the zone of*
*bioturbation determined by XRD (top 12 cm).*

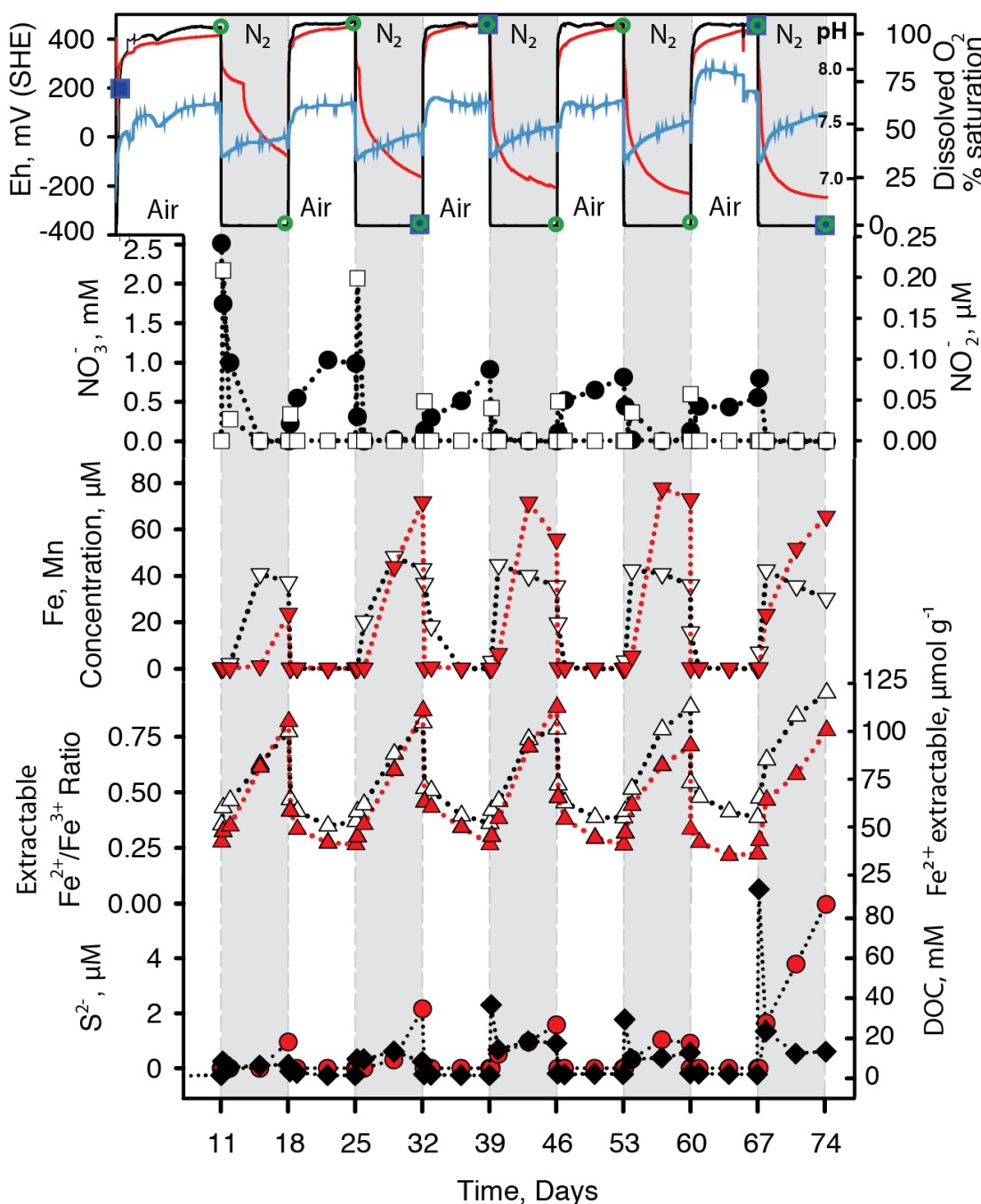


Figure 3: Aqueous chemistry and iron extraction data with time during reactor
experiments: Solid red line = $E_h$, solid black line = DO, solid blue line = pH, full
black circles = $NO_3^-$, white squares = $NO_2^-$, inverted red triangles = $Fe_{(aq)}$, inverted
white triangles = $Mn_{(aq)}$, red triangles = $Fe^{2+}$ 0.5M HCl extractable, white triangles

*= $Fe^{2+}/Fe^{3+}$ ratio in 0.5M HCl extract, red circles = $S_2^-$, black diamonds = DOC.*
*Sampling points for $^{31}P$ NMR and extracellular enzyme assays (EEA) are shown on*
*the $E_h$ curve ($^{31}P$ NMR = open blue squares, EEA = open green circles).*

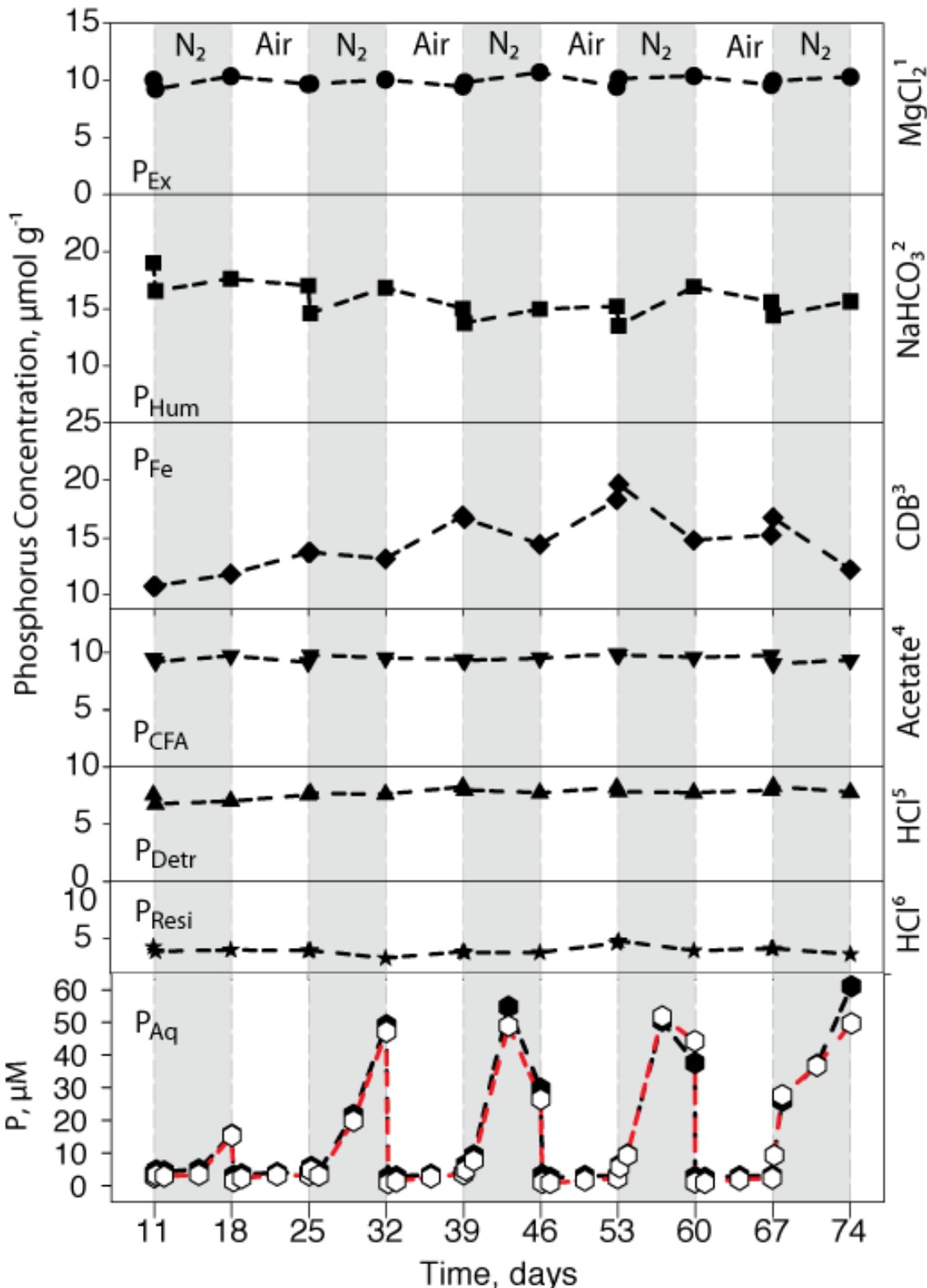

*Figure 4: Aqueous and solid phase phosphorus speciation from sequential*
*chemical extractions with time during the reactor experiment. White panels*
*correspond to time periods with air sparging, grey panels correspond to time*
*periods with $N_2:CO_2$ sparging. Black symbols = total P concentration, white*
*symbols = SRP concentration.*

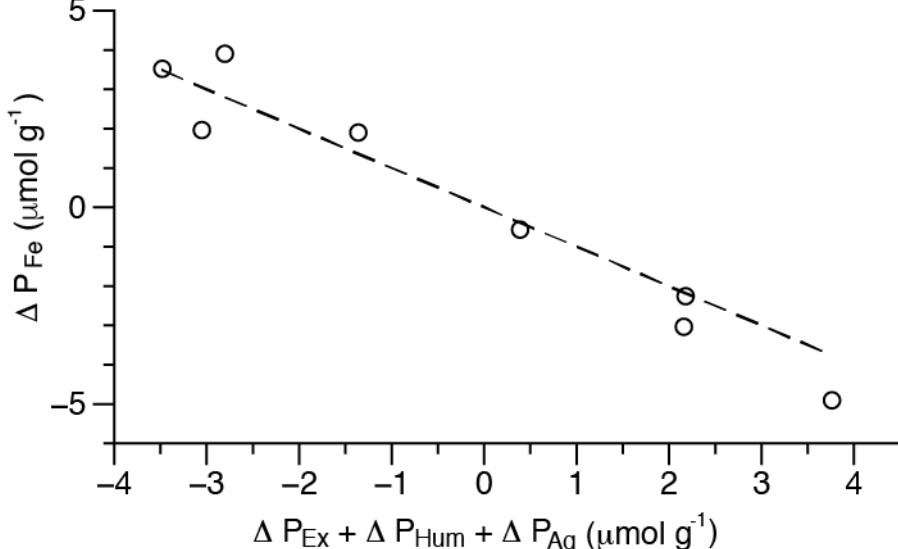


*Figure 5: Change in P distribution between the start and end of each oxic and*
*anoxic period (7 day change). Iron-bound P ($P_{Fe}$) appears to be reversibly*
*redistributed to the loosely sorbed ($P_{Ex}$), humic bound ($P_{Hum}$) and aqueous*
*fractions ($P_{Aq}$). The dashed line is 1:1. Linear regression of the data results in an*
*$R^2$ of 0.95, a slope of -1.1 and p < 0.0001).*





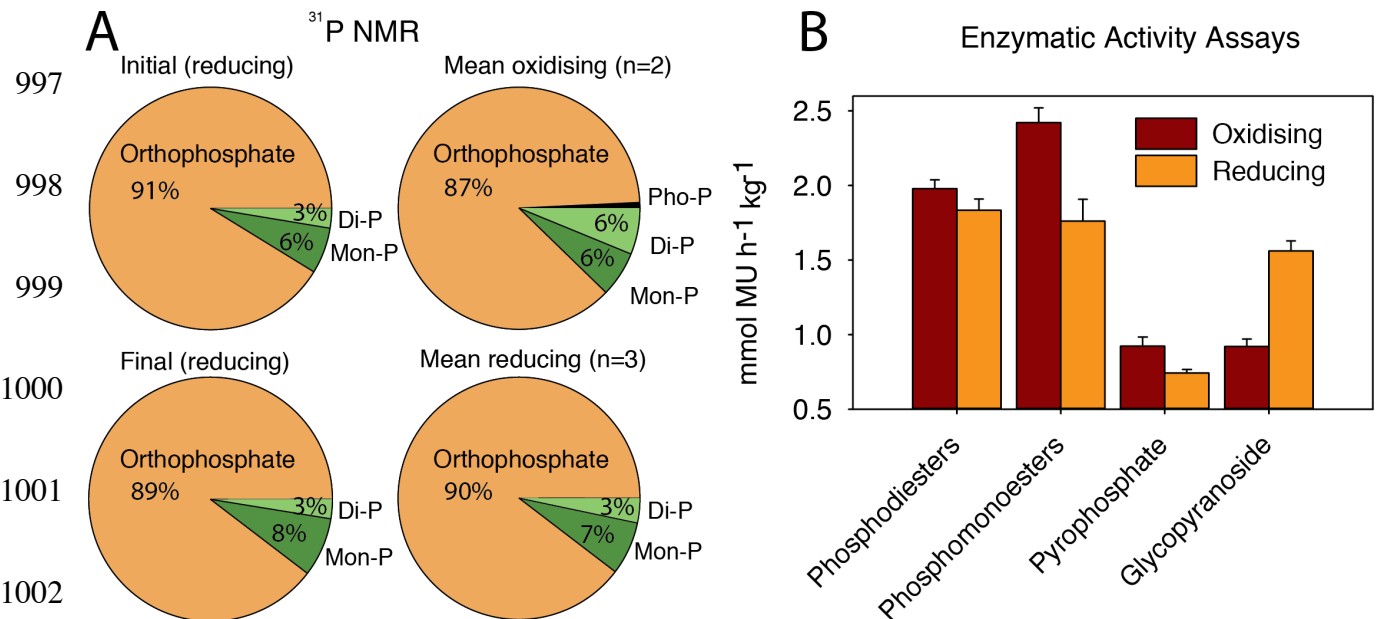







*Figure 6: A) P speciation determined by $^{31}$P NMR for the initial suspension (top*
*left), the final suspension (bottom left), the average of samples from oxic*
*conditions n=2 (top right), the average of samples from anoxic conditions n=3*
*(bottom right). Pho-P = phosphonates, Di-P = diester-P, Mon-P = monoester-P.*
*Polyphosphate was not detected at concentrations >1% in any of the samples*
*analyzed. B) Average extracellular enzyme activities under oxic and anoxic*
*conditions for MUP, DiMUP, PYRO-P and MUGb (n=5)*