# Peer review of "Sediment phosphorus speciation and mobility under"

_Biogeosciences, 2016_

## Referee Comment (RC1) · Anonymous Referee #1 · 14 Feb 2017

GENERAL COMMENTS:

The study by Parsons et al. aims at investigating the effect of the redox potential on sediment phosphorus speciation and resulting changes in soluble phosphate. For mimicking fluctuating redox conditions, the study employs a bioreactor experiment with repeated cycles of anoxia/re-oxygenation by purging with suitable gas mixtures. The study confirms that anoxic conditions result in reductive dissolution of iron(oxyhydr)oxides in the sediment, which in turn results in a subsequent release of iron-bound phosphorus. However, the general approach as well as the finding of an increase in soluble phosphate and decrease in iron-bound P upon reduction of Fe(III) is far from being novel. The concept that the release of phosphorus from anoxic sediment can be attributed to the reduction of a FeOOH-phosphate complex can be traced back to a proposal by Einsele, which was later adapted by Mortimer (Einsele, 1936;

Mortimer 1941, 1942). Hence, the interesting aspect of this study is rather to try to elucidate the redistribution of released P between other P-bearing phases. The main point of criticism of this manuscript is that the novel aspect is not sufficiently emphasized - and, more importantly, not sufficiently discussed in detail.

Another example for the lack of clarity regarding the main message is the listing of the particular aims in the introduction. According to this list, the aims focus on determining (i) polyphosphate cycling; (ii) accumulation of autochthonous $P_o$ species; and (iii) rates of $P_o$ degradation. However, (i) data regarding the aims are visually present in one of six figures (no tables in the manuscript); (ii) the term 'polyphosphate' does not appear in any of the figures; (iii) polyphosphate accumulation was not confirmed and, more importantly, (iv) determining the accumulation of autochthonous $P_o$ species was not possible because "P contribution from individual algal additions ( 1.5 $\mu$mol P g-1) was relatively small compared to the total P mass in the reactor (61 $\mu$mol P g-1) and within the margin of analytical error associated with solid extractions" (p 17, l 373-375)!

Hence, the experimental design used did not match the research aim. Further, fractions other than $P_{Fe}$ which were found to display clear and significant trends over time were $P_{Ex}$ and $P_{Hum}$. Although it was not listed as being a focus of the study presented, the manuscript in fact focuses on these two fractions (see e.g. figures 4 and 5). However, the discussion fails to substantially address the chemical composition and nature of these two fractions and fails to describe related reaction mechanisms.

In view of the above and other significant shortcomings including weaknesses of the manuscript structure/organization, writing style, documentation and technical aspects (see e.g. the first comment below), I would not recommend Biogeosciences to publish the manuscript.

SPECIFIC COMMENTS:

1) To me the description of the operationally defined fractions and corresponding extraction conditions is insufficient and misleading. For example, the abbreviation $P_{CFA}$

is used for "CaCO$_3$ bound P" (p. 10, lines 198-203) determined using the SEDEX acetate buffer. CFA actually stands for carbonate-fluorapatite. The SEDEX method focuses on the determination of carbonate-fluorapatite. However, the term 'carbonate-fluorapatite' is not mentioned anywhere in the manuscript. Why did the authors not use the original fraction descriptions of the SEDEX protocol? Do the authors assume that CaCO$_3$ bound P plays a significant role at the field site or under the experimental conditions? There may be sufficient calcite, however the P$_{CFA}$ fraction remains constant throughout the experiment despite large changes in P$_{Aq}$ (Fig. 4; even detrital apatite shows more variability). This does not point to CaCO$_3$ bound P, this points to actual calcium phosphates because of their slow precipitation/dissolution kinetics under the experimental conditions.

2) Some available data sets are not presented / not discussed although these data sets could be useful to the reader. For example, the authors "postulate that solubility changes to humified DOC are driven by minor (+/- 0.3) pH changes between oxic an anoxic conditions..." yet recorded changes in pH are not presented (p. 16, lines 339-342). Another example are the collected ICP-OES data and the ion chromatography data. The majority of these data is not shown. Calcium and magnesium, for example, would be interesting ions for estimating the saturation state with respect to specific P species (together with the pH) or the inhibition of nucleation of P mineral phases.

3) The ionic strength together with the pH are essential variables in determining saturation indices and adsorption mechanisms. The ionic strength is mentioned once in the manuscript, but corresponding data are not shown (p. 7, lines 130-131: "Surface water was used, rather than distilled water, to provide background ionic strength and avoid osmotic shock to the microbial community."). For the reader, the only possibility to gather information on the ionic strength are the terms "freshwater sediment" and "freshwater marsh" - which is insufficient.

4) In a broader sense, the study is intended to provide guidance in dealing with increased P loadings or for the "management of WWTP effluent" (e.g. p. 22-23). The

experimental design has several limitations with resulting limitations in its applicability to field conditions, which include the given temperature (25°C) and light conditions (dark) and the sediment pH (7.2-7.5) and the ionic strength (freshwater), only to name a few. Such limitations should be mentioned if management recommendations are given.

TECHNICAL ISSUES:

line 64: Which phases are meant here in detail? ("these phases" in the middle of the introduction seems to refer back to a sentence in the abstract)

lines 70-72: I suggest to explain the coupling of these biogeochemical cycles because they are suggested "to play important and complex roles in phosphorus mobility".

line 90: missing comma; I would avoid brackets inside brackets here – and at other places in the manuscript.

line 91: I suggest citing the original source of the methods used, e.g. Ruttenberg.

lines 103-104: There are nested sentences which can be avoided. See also e.g. lines 107-110 and lines 166-169.

lines 119-121: Why aren't the analyzed elements listed in full?

lines 123-124: Which software was used for phase identification?

lines 177-180: 'Method detection limit' has been abbreviated before (e.g. line 174)

line 206: "Li et (Li et al., 2015) al"

line 252: Grammar

lines 379-381 and lines 392-395: Vague and poor sentences

line 396: Unclear structure of headings (e.g. this is placed after "Fe:P ratios". To me, this belongs to "sequential chemical extractions and solid phase P partitioning")

line 407-410: Vague sentence; in what respect "environmentally relevant"?

Section "hydrolytic enzyme activities": This section deals with Figure 6. However, a reference to Figure 6 does not appear anywhere in this section. Figure 6 is split into a and b. It may make sense to include a,b,... in references.

FIGURE 2: It may make sense to label the depth interval used for the bioreactor experiment in b) and to be more consistent in using labels in general (e.g. "Figure 1: A)", "Figure 2: a)"

FIGURE 3: Inconsistent labelling; some axis labels shown some not (e.g. Mn)

FIGURE 4: Did the authors analyze the 'control composition' for the fraction composition before equilibration?

FIGURE 4: Inconsistent axis labels for $P_{Aq}$

FIGURE 6: Font size is too small in A); missing label for polyphosphate

[Figure]

---

## Referee Comment (RC2) · Anonymous Referee #2 · 19 Feb 2017

Parsons et al. studied speciation and mobilization of sediment P under fluctuating redox conditions using a bioreactor for the duration of 72 days. They reported that the mobility of P was controlled mainly by iron oxides and release of P to the aqueous phase occurred after completion of nitrate depletion. Remobilization of P into aquatic phase was limited since higher proportion of released P at anoxic conditions re-distributed among different P fractions. Mineralization of organic P was observed in the oxic condition where the activity of phosphomonoesterases was ∼37% higher than that in the anoxic condition. Accumulation of polyphosphate was not observed.

I believe their study is valuable for understanding how the shot-term fluctuating redox condition affects the P sorption/release mechanisms in relation to other chemical/physical elements, such as pH, Fe, Mn, C, N and S. The manuscript is well written, however, their aims of study (Line 79-87) don't match their major findings. Even though

their aims were focusing on organic P cycling, their main findings were not likely related to organic P cycling, but the inorganic P sorption/desorption mechanisms related to redox chemistry. Their implications didn't include anything related to organic P cycling. Besides, I am not certain if their experimental design (individual algal additions) was reasonable to represent their study site for organic P cycling. Therefore, I recommend that the aims of the study be re-written. The paper is likely to be publishable, but requires some clarification and more detailed explanations in some parts.

A clarification about the extra step (1 M sodium bicarbonate, NaHCO3) the authors added to the SEDEX referred to the paper published by Baldwin (Line 204) is required since this P fraction is the most important fraction in their study (Line 413). The authors indicated that this extra step differentiates between P associated with metal-OM bridging complexes and truly reducible oxide associated P (Line 204). The authors called the P fraction extracted with the sodium bicarbonate as "Humic bound P, PHUM" (Line 208). However, the target P phase extracted with the sodium bicarbonate was not shown in the Baldwin paper. Baldwin observed that sodium carbonate removed highly colored extract, which indicates removal of part of OM. Thus, he assumed the sodium bicarbonate may have removed some OM associated P, however, he didn't mention anything about P associated with metal-OM bridging complexes (HMEP). Baldwin's sediment samples were rich in OM, whereas the sediment sample the authors used in their experiment was not rich in OM. If the authors observed colored extract as Baldwin did, I suggest that they mention it in the manuscript. I am not certain if it is applicable to express the P phase extracted by the sodium bicarbonate as "Humic bound P = HMEP" also due to the following reasons; 1) Li et al. (2015) reported that HMEP could be more recalcitrant than pure mineral Fe-oxides. If so, it is not likely extracted by the sodium bicarbonate. 2) Gerke (2010) reported that HMEP cannot be differentiated by fractionation methods, however, the formation of HMEP could be confirmed by measuring Fe content in the materials; comparison of pyroP extractable [Fe] and acid oxalate extractable [Fe] by McKegue (1967). These measurements may strengthen the authors statement.Âă 3) It is well-known that in the Hedley's sequential fractionation,

0.5 M NaHCO3 (pH 8.5, 16 h shaking time) extracts some organic P (Hedley, White, & Nye, 1982). Since the sodium bicarbonate the authors used (1.0 M NaHCO3) is similar to the Hedley's extractant, it should extract some organic P as well as organic matter (OM), if the shaken time was long enough (the authors didn't show either the shaking time or pH of the extractant). Therefore, OM associated P may be organic P directly associated with OM. 4) The sodium bicarbonate extractant is also similar to the extractant used in the Olsen P test (0.5 M NaHCO3, pH 8.5, 30 min shaking time). The Olsen P extractant should extract labile Mg-Ps and Ca-Ps, such as monetite and brushite (Olsen, Cole, & Watanabe, 1954). Since the sediment sample used in the authors experiment had relatively high proportions of dolomite (8%) and calcite (19%), Ca bound P was expected to be a major form of P. However, PCFA was only 15% of total P present in their sample. Therefore, I assume PHUM possibly includes labile Ca-P minerals. It would be useful to measure the concentration of Ca and Fe in each extract to confirm this. 5) Even though the sorption sites in clays decrease with increasing pH, P sorption onto illite would occur at neutral pH (Manning & Goldberg, 1996). If pH goes toward alkalinity, P adsorbed by illite would be desorbed (Manning & Goldberg, 1996). The sediment sample used in the authors experiment was relatively high in illite (30%), therefore PHUM possibly includes P adsorbed by illite. I assume that the fraction most likely included various inorganic P forms including P associated Ca, Mg and/or clays as well as OM rather than HMEP. However, if the authors are able to explain/show that the bicarbonate extractant is able to remove HMEP, I think this step is crucial in the SEDEX and this should be one of their remarkable findings.

Even though the authors mentioned Fe/Mn-oxides in their abstract, discussion was made only for Fe-oxides but nothing for Mn-oxides.

The authors characterized forms of organic P in the sediment sample by solution 31P NMR spectroscopy and analyze the rate of mineralization of organic P by various enzymatic approaches. The proportion of organic P in the sediment was found to be ~9 to 16 % with the increasing order of; monoesters > diesters > polyphosphate, which

was equivalent of the enzyme activities. The accumulation of polyphosphate was not observed in their experiment settings. (1) Accumulation of polyphosphate or/and pyrophosphate is known to occur as a luxury uptake in algae and microbes (Hupfer, Gloess, & Grossart, 2007) and is often observed in the surface sediments (Giles, Cade-Menun, & Hill, 2011; Hupfer et al., 2007; Jorgensen, Inglett, Jensen, Reitzel, & Reddy, 2015; Li et al., 2015), however, the accumulation of poly- or/and pyrophosphate was not observed in their experiment. The authors commented this was because the experimental condition was not set in excess P (Line 373), despite the actual sampling site being consistently high in primary productivity (Line 369). I am not certain if the conclusion of the polyphosphate cycle in their experimental setting was valid. (2) Some peaks, especially at chemical shifts assigned as alpha- and beta-glycerophosphates, appearing in the monoester regions often belong to phosphodiesters (Doolette, Smernik, & Dougherty, 2009; Jorgensen et al., 2015; Paraskova et al., 2014; Turner, Mahieu, & Condron, 2003). It is important to consider re-calculating these peaks when comparing the ratio of monoesters to diesters, if the authors didn't do it. (3) The recovery rate of total P by the NaOH-EDTA was not shown in the manuscript.

Below are some minor comments/suggestions:

Line 153: An explanation of the reason why the setting (i.e. temperature at 25 C and the dark setting) was chosen should be noted.

Line 192-194: I suggest moving these sentences to introduction or discussion sections.

Line 198: It is not clear which sediment profile was used for the SEDEX (I imagine it was the surface sediment; 0-4 cm). The number of samples used for the SEDEX should be noted. It would be useful to include a table showing a brief method (i.e. each extractant, pH, shaking time, number of wash, etc.) for readers, since the SEDEX method used in their experiment was modified.

Tables for basic physical/chemical characteristics of (1) sediment sample, such as pH, texture, TP, NaOH-EDTA extractable P, OM, N and other elements (i.e. Ca, Fe, Al, Mn,

etc.) and (2) surface water sample such as pH and chemical contents would be useful to readers.

Line 209: PCFA should be expressed as Ca-bound P since the fraction includes not only CaCO3-associated P, but authigenic carbonate fluorapatite and biogenic apatite. Ruttenberg (1992) reported that the first step (MgCl2) can extract ~25% of biogenic CaCO3 (i.e. loosely sorbed P onto CaCO3).

Line 339: The pH data obtained during the experiment during anoxic and oxic states would be useful for readers.

— References: Doolette, A. L., Smernik, R. J., & Dougherty, W. J. (2009). Spiking Improved Solution Phosphorus-31 Nuclear Magnetic REsonance Idintification of Soil Phosphorus Compounds. Soil Sci. Soc. Am. J., 73, 919-927. Gerke J. (2010) Humic (Organic matter)-Al(Fe)-Phosphate Complexes: An Underestimated Phosphate Form in Soils and Source of Plant-Available Phosphate. Soil Sci., 175, 417-425. Giles, C. D., Cade-Menun, B. J., & Hill, J. E. (2011). The inositol phosphates in soils and manures: Abundance, cycling, and measurement. Can. J. Soil. Sci., 91, 397. Hedley, M. J., White, R. E., & Nye, P. H. (1982). Plant-Induced Changes in the Rhizosphere of Rape (Brassica napus Var. Emerald) Seedlings. III. Changes in L Value, Soil Phosphate Fractions and Phosphatase Activity. New Phytol, 91(1), 45-56. Hupfer, M., Gloess, S., & Grossart, H. P. (2007). Polyphosphate-accumulating microorganisms in aquatic sediments. Aquat. Microb. Ecol., 47(3), 299-311. Jorgensen, C., Inglett, K. S., Jensen, H. S., Reitzel, K., & Reddy, K. R. (2015). Characterization of biogenic phosphorus in outflow water from constructed wetlands. Geoderma, 257-258, 58-66. Li, W., Joshi, S. R., Hou, G., Burdige, D. J., Sparks, D. L., & Jaisi, D. P. (2015). Characterizing Phosphorus Speciation of Chesapeake Bay Sediments Using Chemical Extraction, 31P NMR, and X-ray Absorption Fine Structure Spectroscopy. Environ. Sci. Technol, 49, 203-211. McKeague J. A. (1967) An evaluation of 0.1 M pyrophosphate and pyrophosphate-dithionite in comparison with oxalate as extractants of the accumulation products in Podzols and some other soils. Can. J. Soil Sci., 47, 95-99. Manning, B. A., & Goldberg, S. (1996). Modeling arsenate competitive adsorption on kaolinite, montmorllonite and illite. Clays Clay Minerals, 44(5), 609-623. Olsen, R., Cole, C. V., & Watanabe, F. S. (1954). Estimation of Available Phosphorus in Soils by Extraction With Sodium Bicarbonate. Circular 939, U.S. Dept. of Agriculture, 939. Paraskova, J. V., Jorgensen, C., Reitzel, K., Pettersson, J., Rydin, E., & Sjoberg, P. J. R. (2014). Speciation of inositol phosphates in lake sediments by ion-exchange chromatography coupled with mass spectrometry, inductively coupled plasma atomic emission spectroscopy, and 31P NMR spectroscopy. Analy chem., 87, 2672-2677. Turner, B. L., Mahieu, N., & Condron, L. M. (2003). Phosphorus-31 nuclear magnetic resonance spectral assignments of phosphorus compounds in soil NaOH-EDTA extracts. Soil Sci. Soc. Am. J., 67, 497-510.

---

## Author Comment (AC1) · 6 Mar 2017

Response to Anonymous Referee #1 regarding their review of "Sediment phosphorus speciation and mobility under dynamic redox conditions", which was published on February 14th 2017

We thank the referee for her/his thorough and critical review of our manuscript. The comments highlight the need to improve the clarity of the manuscript in places, particularly with respect to emphasizing the key findings, novelty and importance of our work. We agree with the majority of the reviewer's general and specific technical comments and suggestions, and are able to address them as detailed below. For those comments with which we disagree, in particular regarding differences between experimental and field conditions, we provide more supporting information about the in situ conditions at

the sampling site.

General comments:

"…the general approach as well as the finding of an increase in soluble phosphate and decrease in iron-bound P upon reduction of Fe(III) is far from being novel. The concept that the release of phosphorus from anoxic sediment can be attributed to the reduction of a FeOOH-phosphate complex can be traced back to a proposal by Einsele, which was later adapted by Mortimer (1941, 1942). Hence, the interesting aspect of this study is rather to try and elucidate the redistribution of released P between other P-bearing phases."

We agree that the finding of aqueous phosphorus release from sediments under anoxic conditions is in itself well established. We acknowledge this at the beginning of the manuscript, on lines 60-66, and indeed, cite Mortimer (1941) and other seminal works within this section. The novelty and relevance of the current study is, we believe, more extensive than highlighted by the reviewer. Specifically, the novelty of our work includes the following.

1. Acquisition of a comprehensive data set that describes the fully mass balanced redistribution of P between different solid and aqueous sediment pools as a function of changes in the redox state of the system. Many previous studies have shown aqueous P release from sediments under anoxic conditions and some have established the origin of the aqueous P, e.g. organic necromass (Joshi et al., 2015), microbial polyphosphates (Hupfer et al., 2007), or mineral bound P (Petticrew and Arocena, 2001). However, no previous studies have quantitatively investigated the redistribution of P among the mineral and organic fractions of a sediment during redox fluctuations. Most importantly, we demonstrate that only a very small proportion (4.5%) of P associated with reducible iron(III) (oxy)hydroxides actually ends up in the aqueous phase upon reductive dissolution of the iron(III) (oxy)hydroxides, even in carbonate-buffered sediments under slightly alkaline pH where P sorption is considered to be relatively

ineffective. Further, we demonstrate that polyphosphate accumulation and release are not major P cycling processes within this freshwater sediment (as reported previously in other redox oscillating environments). 2. Use of a controlled reactor system to simulate repetitive cycling between oxic and anoxic conditions. Oscillating fluctuations of redox conditions are ubiquitous in shallow sediments but are rarely investigated in laboratory experiments. Many experiments simulate a single oxidation or reduction event (e.g., Matisoff et al., 2016). To the best of our knowledge, the effects of successive oxic-anoxic cycles on P speciation, mineral association and mobility in freshwater sediments have not previously been investigated. The approach presented offers the possibility to determine cumulative effects of redox cycles, and to establish which (im)mobilisation processes are reversible. 3. Delineation of the interplay between redox conditions, mineralogical changes and activities of hydrolytic phosphatases. We demonstrate, for the first time, that phosphatase activities vary systematically with redox conditions, with higher activities occurring under oxidising conditions and lower activities occurring under reducing conditions. This trend appears to be specific to phosphatase enzymes: a hydrolytic enzyme of the cellulose degradation pathway shows the exact opposite trend. We propose that this observation reflects changes in enzyme production by the microbial community in response to phosphate scarcity during oxic conditions, because of sorption of phosphate to iron(III) (oxy)hydroxide minerals.

We agree that the above key points need to be more forcefully stated in the revised version of the manuscript.

"Another example for the lack of clarity regarding the main message is the listing of the particular aims in the introduction. According to this list, the aims focus on determining (i) polyphosphate cycling; (ii) accumulation of autochthonous Po species; and (iii) rates of Po degradation."

To clarify, our over-arching research aim was stated in the introduction on lines 79-81: "The aim of this study is to elucidate the microbial and geochemical mechanisms of in-sediment phosphorus cycling and release associated with commonly occurring short

redox fluctuations (days) at the SWI in shallow eutrophic environments."

Polyphosphate accumulation due to microbial activity has been proposed as a key process affecting P cycling under oscillating redox conditions in some environments, where polyphosphate can account for up to 10% of total phosphorus (Hupfer et al., 2007). For this reason, polyphosphate accumulation was investigated alongside other possible P cycling mechanisms. The time series 31P NMR analyses demonstrate that polyphosphate accumulation and release are not major processes within our experiment: polyphosphate never accounts for more than 1% of total P.

We do agree with the reviewer that polyphosphate cycling is only one of the many different processes investigated within this research and, especially given its negligible role in our experiment, it should not be highlighted as prominently in the introduction. This will be addressed by rewriting the Introduction of the revised manuscript, which will put more emphasis on the novel aspects listed in response the reviewer's first comment.

". . .(ii) the term 'polyphosphate' does not appear in any of the figures; (iii) polyphosphate accumulation was not confirmed and, more importantly, (iv) determining the accumulation of autochthonous Po species was not possible. . ."

The term polyphosphate is not shown in any of the figures, as it was never present in any of the samples at a concentration greater than 1%, thus, confirming that polyphosphate accumulation did not constitute a major P cycling process within this experiment in contrast to previous studies. Although this is noted within the manuscript on lines 443 to 446, we propose to include the statement "Polyphosphate was not detected at a concentration >1% in any of the samples analysed" within the figure caption for Figure 6 to clarify this point in the revised version of the manuscript.

We agree with the reviewer that accumulation of Po during the experiment due to algal additions is unlikely due to the small amounts of algal matter added and the relatively short experimental timescale (74 days). As pointed out by the reviewer, we state this

on lines 373 to 375 of the manuscript. In addition, we present two lines of experimental evidence suggesting that Po accumulation is not a prominent process within this sediment and that Po degradation is very rapid. These include:

1) The low concentration of Po compared to orthophosphate determined by 31P NMR (9% vs 91%) and through sequential extractions (7% PRes). 2) The results of phosphatase enzyme activities, which show the capacity for rapid Po hydrolysis under both oxidizing and reducing conditions.

We will modify the introduction to shift the focus away from polyphosphate and Po accumulation toward the key novel aspects. We also plan to remove the statement about Po on lines 446-447.

"Hence, the experimental design used did not match the research aim"

The goal of this research, as noted on lines 79-81, was to elucidate the chemical, mineralogical and microbial processes that control redistribution and release of P from the solid phase to the aqueous phase during oscillatory redox conditions. We believe that the experimental design and analytical methods used are appropriate for this aim, as evidenced by:

1) Production of the first fully mass balanced redistribution of P between different sedimentary pools during redox oscillating conditions. 2) Demonstration of reversibility of P transfers between solid and aqueous pools. 3) Assessment of the Po hydrolysis capacity of the sedimentary microbial community through the use of phosphatase enzyme activity assays.

"...the discussion fails to substantially address the chemical composition and nature of these two fractions (PEx and PHum) and fails to describe related reaction mechanisms."

The chemical composition of the PHum pool is discussed in great detail within the manuscript. There is an entire subsection of the discussion devoted to the interpretation of the chemical composition, nature and reaction mechanisms associated with the PHum fraction (lines 396 to 416). If there are specific reactions or alternative interpretations of the PHum pool, which the reviewer believes we have missed within this discussion, we would be keen to include them in a revised version of the manuscript. Additionally, the rationale for the inclusion of the PHum pool within the extraction scheme and with respect to P binding mechanisms – based on the most current literature – is provided on lines 204-207 of the methods section.

Detailed discussion of the chemical composition of each of the phases extracted within the SEDEX method, including PEx, were not included in the manuscript as the method has been broadly used to study P speciation in sediments for approximately 25 years. Detailed descriptions of the phases targeted by the SEDEX extraction scheme, and the reaction mechanisms are provided within the original method by Ruttenberg (1992), which is referenced on line 199 in the methods section and again on line 372 in the discussion of the manuscript. However, to address the concern of the reviewer and to make the manuscript easier to follow for readers unfamiliar with the SEDEX extraction scheme, we will include brief descriptions of each of the pools in the original SEDEX method (i.e., PEx, PFe, PCFA, PDetr and PResi) within the Methods section. We further propose to revise the discussion section by explicitly including probable reaction mechanisms, where appropriate (e.g. adsorption, co-precipitation and the formation of ternary complexes).

Specific comments:

"1) To me, the description of the operationally defined fractions and corresponding extraction conditions is insufficient and misleading"

It was certainly not our intention to mislead readers with the description of the operationally defined fractions targeted by the SEDEX method. As stated above, we will include brief summaries of the chemical composition and P binding mechanisms associated with each pool extracted with the SEDEX method. With regards to the description of the fraction extracted with acetic acid, Ruttenberg (1992) identifies this fraction as "authigenic carbonate fluorapatite + biogenic apatite + CaCO3-associated P". We agree with the reviewer that simply referring to this fraction as "CaCO3 bound P" may be misleading and we will therefore revert to the original definition of Ruttenberg.

"2) Some available data sets are not presented / not discussed although these data sets could be useful to the reader."

We agree with the reviewer's point that more complete data could be useful to the reader. A large amount of additional data was produced during our experimental study, e.g. ICP-OES data from the aqueous fraction, ICP-OES data from the SEDEX method and ion-chromatography data. Not all the data are shown in the manuscript to avoid diluting the key points of the paper. We propose to include pH data within Figure 3, as these data are directly discussed within the manuscript, and to include all additional data sets in tabulated and graphical form within the supplementary material of the revised version of the manuscript. In doing so, the additional data will be available to the reader, but without detracting from the key messages of the project with respect to P cycling. Saturation indices for major P minerals were calculated with the available solution data using PHREEQC, at all time points during the experiment. The results of these calculations will also be included within the supplementary information of the revised version of the manuscript.

"3) The ionic strength together with the pH are essential variables in determining saturation indices and adsorption mechanisms...For the reader, the only possibility to gather information on the ionic strength are the terms "freshwater sediment" and "freshwater marsh" – which is insufficient".

We will include the pH data within Figure 3 and the ionic strength in the "Experimental redox oscillation: Aqueous chemistry section" (294-352). Furthermore, we propose to include the calculated saturation indices for all the major P mineral phases within the supplementary information.

"4) In a broader sense, the study is intended to provide guidance in dealing with increased P loadings or for the "management of WWTP effluent" (e.g. p. 22-23). The experimental design has several limitations with resulting limitations in its applicability to field conditions, which include the given temperature (25oC) and light conditions (dark) and the sediment pH (7.2-7.5) and the ionic strength (freshwater), only to name a few. Such limitations should be mentioned if management recommendations are given."

We fully recognize that there are (always) limitations to extrapolating experimental results obtained in the laboratory to field conditions. Nonetheless, the processes described in our study have, we believe, general implications for the management of internal and external nutrient loads in small lentic systems. Admittedly, referring to WWTP effluent in the implications section may be too specific given the focus of the bioreactor experiment on fundamental processes. We propose to replace the statement about WWTP effluent with a more general one about the role of external nitrate loading in the remobilization of legacy P from bottom sediments.

We also wish to emphasize that the temperature, light conditions, pH and ionic strength in the experiment actually closely resemble those found at the sampling site during summer/early fall. The pH in the experiments is buffered by the calcium carbonate naturally occurring in the sediment and matches values measured in the bottom waters of Cootes Paradise (see Figure 2 and lines 288-290). Similarly, the average temperature at the SWI measured at the sampling location in August 2014 is $23.8°C$, while the absence of light is representative of processes taking place at or below the SWI, in particular because a thick mat of green filamentous algae covers the pond's surface during the growing season (as shown in Figure 1 A). We opted to mimic summer/early fall conditions, because sediments experience intense redox fluctuations during this period due to active bioturbation and pronounced diurnal cycles of photosynthesis/respiration. In addition, this is also the time of greatest organic matter input at the SWI. Thus, we strongly believe that the key processes identified within the reactor experiment represent some of the key biogeochemical processes occurring within the topmost sediment at the field site, during the time of intense benthic activity and exchanges. We recognize that we must make this point more clearly in the revised manuscript

Technical Issues:

"Line 64: Which phases are meant here" – Agreed, this is unclear, we propose replacing "these phases" with "iron(III) (oxy)hydroxides" in the revised version of the manuscript.

"Lines 70-72: I suggest to explain the coupling of these biogeochemical cycles..." Detailed explanations of the importance of these coupled cycles were not provided for brevity. Instead the reader was referred to the references cited. In the revised manuscript a short paragraph will be added to summarize the major known pathways through which of carbon and sulfur affect P cycling.

"Line 90: Missing comma and brackets inside brackets" – Agreed this will be fixed.

"Line 91:...Citing the original source of the methods" – Agreed, we have cited the Ruttenberg method and others within the methods section and discussion but we will definitely add references to the appropriate methods here.

Lines 103-104, 107-110 and 166-169 – We agree with the reviewer, these nested sentences can be avoided and we propose to reword these sections to improve readability.

"Lines 119-121: Why aren't the analyzed elements listed in full" – The full elemental list wasn't provided for brevity. We accept the reviewer's point that this list, and indeed the data, would be useful to readers. We propose to include the complete data for all elements in tabulated and graphical forms in the supplementary information.

"Lines 123-124: Which software was used for phase identification" – The software used was PANalytical's Highscore+. This detail, including the version number will be added to the revised manuscript.

"Lines 177-180: 'Method detection limit' has been abbreviated before" – Agreed.

Line 206: "Li et (Li et al., 2015) al" –This referencing error will be corrected.

"Line 252: Grammar" – Agreed, we propose rewording to "Excitation fluorescence was set at 365 nm and emission intensity at 450nm was recorded at 5 minute intervals over a 6-hour period".

"Lines 379-381 and lines 392-395: Vague and poor sentences" – We agree with the reviewer and will rewrite the offending sentences in the revised manuscript.

"Line 396: Unclear structure of headings" – We agree with the reviewer, this makes more sense as a subsection within "Sequential chemical extractions and solid phase P partitioning". This will be corrected in the revised manuscript.

"Line 407-410: Vague sentence; in what respect "environmentally relevant" – We will rephrase this for clarity. The papers cited demonstrate that mixed Fe(III)-OM-phosphate and arsenate complexes can be synthesized in the laboratory. However, to the best of our knowledge, there is no direct evidence for the existence of these complexes in natural environments, largely because of the difficulty of measuring such complexes at low concentrations in structurally mixed and complex natural samples. Nonetheless, the recent spectroscopic evidence of synthesized Fe(III)-OM-phosphate complexes supports the hypothesis that these complexes can also form, and indeed probably form, in natural freshwater environments. We will clarify this point in the revised manuscript.

Section "hydrolytic enzyme activities" – No reference to Figure 6. - We agree with the reviewer: there should be a reference to Figure 6 within this section. This will be corrected in the revised version of the manuscript.

Figure 2: Label depth interval used in the bioreactor – We agree and will add this.

Figure 3: Inconsistent labeling; some axis labels shown some not (e.g. Mn) – We disagree that this is inconsistent. A second axis is only used when two sets of data

cannot be shown effectively on the same scale. For Fe and Mn, the same scale can be used and a common "concentration" label is applied. This is clearly explained in the figure caption. We do not intend to change the figure labeling.

Figure 4: P fractionation before equilibration is not shown – The objective of the experiment was to determine the redistribution of P cycling during redox oscillating conditions, therefore, the first sample analyzed was the one immediately starting the redox oscillations (i.e., the experimental time origin).

Figure 4: Inconsistent axis labels for PAq - We use both $\mu$M and mg P L-1 as some readers may be more familiar with one unit or the other. We do not propose to make any changes.

Figure 6: Font size, missing polyphosphate label – Font size will be increased. There is no polyphosphate label as polyphosphate was below the detection limit. This information will be included in the caption of Figure 6 as follows: "Polyphosphate was not detected at a concentrations >1% in any of the samples analyzed".

References:

Hupfer, M., Gloess, S., Grossart, H., 2007. Polyphosphate-accumulating microorganisms in aquatic sediments. Aquat. Microb. Ecol. 47, 299–311. doi:10.3354/ame047299 Joshi, S.R., Kukkadapu, R.K., Burdige, D.J., Bowden, M.E., Sparks, D.L., Jaisi, D.P., 2015. Organic Matter Remineralization Predominates Phosphorus Cycling in the Mid-Bay Sediments in the Chesapeake Bay. Environ. Sci. Technol. 49, 5887–5896. doi:10.1021/es5059617 Matisoff, G., Kaltenberg, E.M., Steely, R.L., Hummel, S.K., Seo, J., Gibbons, K.J., Bridgeman, T.B., Seo, Y., Behbahani, M., James, W.F., Johnson, L.T., Doan, P., Dittrich, M., Evans, M.A., Chaffin, J.D., 2016. Internal loading of phosphorus in western Lake Erie. J. Gt. Lakes Res. 42, 775–788. doi:10.1016/j.jglr.2016.04.004 Petticrew, E., Arocena, J., 2001. Evaluation of iron-phosphate as a source of internal lake phosphorus loadings. Sci. Total Environ. 266, 87–93. doi:10.1016/S0048-9697(00)00756-7

Please also note the supplement to this comment:
http://www.biogeosciences-discuss.net/bg-2016-533/bg-2016-533-AC1-supplement.pdf

―――――――――――――――――

---

## Author Comment (AC2) · 6 Mar 2017

Response to Anonymous Referee #2 regarding their review of "Sediment phosphorus speciation and mobility under dynamic redox conditions", which was published on February 19th 2017

We thank the referee for his/her careful and thorough review of our manuscript. We appreciate their very detailed comments particularly considering interpretation of the sequential extraction and 31P NMR results. The references and comments provided will allow us to significantly improve the quality of the final manuscript. We agree with all of the referee's comments and will take actions to address their concerns in a revised version of the manuscript.

General comments:

"The manuscript is well written, however, their aims of study (Line 79-87) don't match their major findings. Even though their aims were focusing on organic P cycling, their main findings were not likely related to organic P cycling, but the inorganic P sorption/desorption mechanisms related to redox chemistry. Their implications didn't include anything related to organic P cycling. Besides, I am not certain if their experimental design (individual algal additions) was reasonable to represent their study site for organic P cycling. Therefore, I recommend that the aims of the study be re-written."

We agree with the reviewer, although we believe that lines 79-81 (repeated below) are a good reflection of the aims of the project, the list of particular aims on lines 81-87 do not match the most novel findings from this study. We will re-write this section to better reflect the key points from experimental results in the revised version of the manuscript.

"The aim of this study is to elucidate the microbial and geochemical mechanisms of in sediment phosphorus cycling and release associated with commonly occurring short redox fluctuations (days) at the SWI in shallow eutrophic environments."

Detailed comments/suggestions:

Clarification of the additional step in the extraction protocol (1M NaHCO3) –

We think that the reviewer has raised an important point, which we agree requires further clarification in the manuscript. The 1M NaHCO3 extract was mildly brown in color and we will definitely include that observation in the revised version of the manuscript. We agree that our initial assertion that this extraction corresponds uniquely to humic-metal ternary complexes is likely over simplification/over interpretation. Ultimately, of course this is an operationally defined P fraction, but we agree that further discussion of what this fraction represents would be useful, particularly as it accounts for such a large proportion of TP. We agree that this fraction likely includes some Ca-P and Mg-P as well as some organic-P. We did in fact measure Ca, Mg, Mn and Fe on each extract and we propose to include these results within the supporting information of the manuscript. We also measured SRP and TDP on the extracts but chose

not to include the SRP results, as we do not feel confident that the initial SRP/TP ratio in the solid phase would be preserved during the extractions. If the reviewer feels that this information would be useful to readers we would be happy to include it within the main manuscript. We believe that references provided by the reviewer, as well as their insightful comments within their review, will allow us to improve this aspect of the manuscript in a revised version. We will also include the reaction time for each extract within the supplementary material to allow more direct comparison to extracts such as Olsen P and Hedley's.

Even though the authors mentioned Fe/Mn-oxides in their abstract, discussion was made only for Fe-oxides but nothing for Mn-oxides

We agree, this is an oversight. We did include measurements of Mn in the aqueous phase (Figure 3) and for each of the extracts but this data was not presented or discussed in the manuscript. We propose to include the Mn extract data in the supporting information, and extend the discussion to include extracted Ca, Mg, Mn and Fe for the key extraction steps.

Accumulation of polyphosphate or/and pyrophosphate is known to occur as a luxury uptake in algae and microbes (Hupfer, Gloess, & Grossart, 2007) and is often observed in the surface sediments (Giles, CadeMenun, & Hill, 2011; Hupfer et al., 2007; Jorgensen, Inglett, Jensen, Reitzel, & Reddy, 2015; Li et al., 2015), however, the accumulation of poly- or/and pyrophosphate was not observed in their experiment. The authors commented this was because the experimental condition was not set in excess P (Line 373), despite the actual sampling site being consistently high in primary productivity (Line 369). I am not certain if the conclusion of the polyphosphate cycle in their experimental setting was valid.

The reviewer has highlighted the need to clarify the text in regard to this point. The TP concentrations in the sediments at the field site are high (Bowman and Theysmeyer, 2014), and certainly contribute significantly to the high primary productivity observed

at the site via internal loading (Chow-Fraser et al., 1998) due in part to variable re-dox conditions. Consequently, the TP concentrations in the reactor are high at all times throughout the experiment. The statement on line 373 refers only to the fact that the algal matter added during the experiment, as a source of organic carbon to fuel metabolic processes, did not represent a major contribution of phosphorus to the system. This should not be taken to mean that P is highly limiting in the system at all times. As TDP concentration is very high during anoxic conditions and very low during oxic conditions, luxury uptake required for polyphosphate production, which is known to occur under oxic conditions, cannot occur. This is despite very high TP concentrations, periodically high TDP concentrations and oscillatory redox conditions. We believe that this observation of the competition between luxury microbial uptake and immobilisation within redox sensitive minerals is a valuable contribution to the literature.

(2) Some peaks, especially at chemical shifts assigned as alpha- and beta-glycerophosphates, appearing in the monoester regions often belong to phosphodi-esters (Doolette, Smernik, & Dougherty, 2009; Jorgensen et al., 2015; Paraskova et al., 2014; Turner, Mahieu, & Condron, 2003). It is important to consider re-calculating these peaks when comparing the ratio of monoesters to diesters, if the authors didn't do it.

We did not apply this correction but will do so in the revised version of the manuscript. We thank the reviewer for bringing this to our attention.

(3) The recovery rate of total P by the NaOH-EDTA was not shown in the manuscript.

We apologise for this omission, the recovery will be included in the revised version of the manuscript.

Line 153: An explanation of the reason why the setting (i.e. temperature at 25 C and the dark setting) was chosen should be noted.

We agree with the reviewer. The average temperature at the SWI at the sampling loca-

tion averaged 23.8oC in August 2014, we chose 25oC as it is close to field values, at least for summer conditions, and allowed for rapid biogeochemical cycling, which was the focus of this study. Dark conditions were chosen as the conditions were designed to simulate cycling in the top 12cm of sediment, which are rapidly turned over due to bioturbation. Dark conditions prevail below the surface at all times and during the growing season at the sediment-water interface due to the growth of thick algal mats on the water surface. We will include this rationale in the revised version of the manuscript.

Line 192-194: I suggest moving these sentences to introduction or discussion sections.

We agree with the reviewer, this sentence will be moved to the discussion in the revised version of the manuscript.

Line 198: It is not clear which sediment profile was used for the SEDEX (I imagine it was the surface sediment; 0-4 cm). The number of samples used for the SEDEX should be noted. It would be useful to include a table showing a brief method (i.e. each extractant, pH, shaking time, number of wash, etc.) for readers, since the SEDEX method used in their experiment was modified.

We apologise for this lack of clarity. The SEDEX protocol was applied to solid phase samples taken from the reactor during the laboratory experiment. The sediment used in the reactor was sampled from 0-12cm sediment depth in the field (as stated on line 129). A duplicate of each sample was analysed through SEDEX, the average of these two samples is presented. We will include these details and reword this section for improved clarity in the revised version of the manuscript. We will additionally include a table, as suggested by the reviewer, detailing the extraction procedure.

Tables for basic physical/chemical characteristics of (1) sediment sample, such as pH, texture, TP, NaOH-EDTA extractable P, OM, N and other elements (i.e. Ca, Fe, Al, Mn, and (2) surface water sample such as pH and chemical contents would be useful to readers.

Agreed, we will include these tables in the revised manuscript.

Line 209: PCFA should be expressed as Ca-bound P since the fraction includes not only CaCO3-associated P, but authigenic carbonate fluorapatite and biogenic apatite. Ruttenberg (1992) reported that the first step (MgCl2) can extract âĹij25% of biogenic CaCO3 (i.e. loosely sorbed P onto CaCO3).

We agree with the reviewer. We propose to revert to the name applied to this extraction by Ruttenberg in the original method.

Line 339: The pH data obtained during the experiment during anoxic and oxic states would be useful for readers.

We agree and propose to include time series pH data within Figure 3.

References:

Bowman, J., Theysmeyer, T., 2014. 2013 RBG Marsh Sediment Quality Assessment (No. Report No. 2014-14). Royal Botanical Gardens, Burlington, Ontario. Chow-Fraser, P., Lougheed, V., Le Thiec, V., Crosbie, B., Simser, L., Lord, J., 1998. Long-term response of the biotic community to fluctuating water levels and changes in water quality in Cootes Paradise Marsh, a degraded coastal wetland of Lake Ontario. Wetl. Ecol. Manag. 6, 19–42. doi:10.1023/A:1008491520668

Please also note the supplement to this comment:
http://www.biogeosciences-discuss.net/bg-2016-533/bg-2016-533-AC2-supplement.pdf

---

## Author Response (AR1)

**Response to the anonymous reviews of "Sediment phosphorus speciation and mobility under dynamic redox conditions"**

We thank both reviewers for their careful consideration and thorough review of our manuscript. Their comments and criticisms have allowed us to significantly improve the quality and clarity of the manuscript in its revised form. As you will see from the marked up manuscript and the table of changes below, we have made substantial revisions to the original submission in response to the reviewers concerns and suggestions. We have addressed all of the concerns highlighted by the reviewers and have made numerous additional changes to improve the readability, structure and technical content of our manuscript.

Briefly we have:

1. Re-written large sections of the manuscript, particularly emphasizing the aims of the study and novelty of the work, which we accept were not well expressed in the first iteration of the manuscript. We have also added a much more detailed discussion of the reactions occurring during the reactor experiment and within the sequential extraction procedure.
2. Included an additional Table, as requested by reviewer 2, to clarify the sequential extraction procedure used.
3. Calculated saturation indices for major Fe, S, Ca and P minerals using PHREEQC.
4. Included the additional data requested by both reviewers, within the manuscript and supplementary information.
5. Modified Figures 2, 3, 4 and 6.

A full list of changes is detailed in the table below:

| Line number | Changes made and rationale |
|---|---|
| Line 15 | Emphasize that our experiments deal with the reactions occurring within shallow sediment rather than directly at the SWI |
| Lines 21-23 | Additional sentence to emphasize the breadth of data presented within the manuscript and one of the key novel contributions (mass balanced redistribution of P within solid and aqueous phases). |
| Lines 26-28 | Provided more detail regarding the redistribution of P within the solid phase (key finding of the manuscript), without over interpretation of the extracted fractions. |
| Lines 29-31 | Added the key results of the $^{31}$P NMR analyses to the abstract to highlight that the paper is mostly about cycling of inorganic P species rather than organic-P. |
| Lines 34-36 | Re-write of the concluding section of the abstract to better reflect the key conclusions of the study. |
| Line 67 | Changed reference list to reflect the original study by Einsele. |

| Lines 72-76 | New short section to highlight the common accumulation of polyphosphate in sediments, particularly under oscillating redox conditions. |
|---|---|
| Lines 80-94 | Brief section added, at the request of reviewer 1, to discuss the key processes coupling the cycles of Fe, S, C and P. |
| Lines 95-99 | Brief section added to highlight the novelty of laboratory experiments, which reflect dynamic, fluctuating natural environments. |
| Lines 102-107 | Re-write of the key aims of the study more appropriate to the study design and to better reflect the novel findings |
| Lines 115-135 | Field site and sampling section – Several parts restructured and nested sentences re-written to improve readability. |
| Line 140 | Statement about sediment elemental concentrations from cores removed as this data is not shown (data lost). |
| Line 142-143 | Details of software used for phase ID and quantitative phase analysis is now provided. |
| Lines 181-184 | Statement added to explain the choice of reactor conditions e.g. temperature, dark etc |
| Lines 184-189 | Re-written section to improve readability |
| Lines 210-212 | Re-phrase to clarify that SEDEX extractions were performed on a time series from the reactor experiment rather than a depth profile. |
| Lines 218-228 | Section re-written in more detail to explain the modifications made to the SEDEX method and to reference the new table. |
| Lines 233-235 | New statement explaining how saturation indices were calculated. |
| Line 237 | New statement to clarify that time series $^{31}P$ NMR data was collected from reactor samples. |
| Lines 278-279 | Re-written sentence for clarity. |
| Lines 321-325 | Re-written to provide more detailed information on pH trends within the reactor and referencing the data, which is now provided in Figure 3. |
| Lines 325-326 | Inclusion of ionic strength data. |
| Line 326-330 | Re-write to clarify which field conditions the reactor is replicating. |
| Lines 343-345 | New statement regarding the effect of the reactor experiment on the biogeochemical function of the sediment. |
| Lines 348-351 | Section moved from methods as suggested by reviewer 2. |
| Lines 355-358 | New statement regarding the results of the thermodynamic model. |
| Line 378 | New reference to Figure 3 added to the reactor pH data. |
| Lines 389-390 | New statement to highlight the availability of all aqueous data in the supporting information. |
| Lines 392-394 | New statement clarifying the relevance that the total extraction matches the sum of the sequential extracts. |
| Lines 396-401 | New discussion about the stability and variability of different P fractions to highlight which fractions are involved in P redistribution during redox cycling. |

| | |
|---|---|
| Lines 411-423 | New discussion about the content of the $P_{Ex}$ fraction, how it compares to previous studies and the reaction mechanisms likely causing changes to $P_{Ex}$ concentration over time. |
| Lines 424-430 | Additional discussion about the probable content of the $P_{Hum}$ fraction, how it compares to other common extractions and the rationale for its inclusion. |
| Lines 431-445 | Re-written previous discussion on the content of the $P_{Hum}$ fraction and comparison to similar commonly used extractions. |
| Lines 446-468 | Re-written section. |
| Lines 469-476 | New discussion on the reaction mechanisms causing release and immobilization of P from and to the $P_{Hum}$ fraction. |
| Lines 477-483 | New discussion of the $P_{Fe}$ fraction – content and reaction mechanisms. |
| Lines 484-488 | New discussion on the stability of the remaining P fractions. |
| Lines 494-495 | Highlighting availability of solid phase chemistry data in the supporting information. |
| Lines 508-510 | Inclusion of results from thermodynamic model. |
| Lines 521 and 526 | References to Figure 6 added. |
| Lines 529-530 | Additional information regarding the interpretation of the enzyme activities. |
| Lines 534-536 | Additional statement to highlight the novelty of the enzyme activity results. |
| Lines 549-551 | |
| Lines 551-555 | Acknowledgement of the importance of degradation of P-diesters during analysis. |
| Lines 564-565 | Additional link made between the $^{31}P$ NMR results and the sequential extraction results. |
| Lines 590-598 | Re-write to be a) less specific about WWTPs, b) acknowledge the limits to the reactor experiment at predicting field scale response. |
| Lines 610-611 | Added acknowledgements to people whose contribution should have been acknowledged in the first version of the manuscript. |
| Table 1 | New table detailing the full sequential extraction scheme used. |
| Figure 2 | We have added the depth interval used in the reactor experiment. |
| Figure 3 | We have removed inconsistencies and improved clarity in the axis labels. |
| Figure 4 | We have removed the $2^{nd}$ axis showing mg P $L^{-1}$ for consistency. |
| Figure 6 | We have added a statement explaining the absence of polyphosphate on the pie charts and increased the size of the text as suggested. |
| Supplementary Material | Full tabulated time series data for the reactor experiment is now provided. |

[revised manuscript text omitted]

Chris Parsons 2017-5-3 12:34

Chris Parsons 2017-4-26 16:42

Chris Parsons 2017-4-26 16:47

Chris Parsons 2017-4-26 16:49

Chris Parsons 2017-4-30 01:45

Chris Parsons 2017-4-26 17:02
**Moved up [2]:** P associated with OM ($P_{Hum}$) .

Chris Parsons 2017-4-30 01:45

Chris Parsons 2017-4-30 03:07

Chris Parsons 2017-4-30 12:39

Chris Parsons 2017-4-30 12:39

Chris Parsons 2017-4-30 12:40

concomitant changes to moisture content and redox conditions which prevents isolation of the causal variable in field investigations (Rezanezhad et al., 2014). Therefore, this is to the best of our knowledge, the first direct demonstration of phosphatase activity changes in response to changing redox conditions. We postulate that under anoxic conditions when phosphorus availability in the aqueous phase is high, production of extracellular phosphatase enzymes by the microbial community is down regulated. Conversely, when bioavailable phosphorus is removed from solution under oxic conditions, extracellular phosphatase production is up regulated in response. Adjustments to enzyme production in response to changes in phosphate availability must occur on short timescales (hours/days) for such trends to be observable during the bioreactor experiment. An inverse relationship between phosphatase activities and phosphate concentration, has previously been shown spatially in wetlands by Kang and Freeman (1999) but to our knowledge never temporally in sediments.

**$^{31}$P NMR**

Results from $^{31}$P NMR analyses (Figure 6A) show that the majority of phosphorus was present in the solid phase as ortho-phosphate (84-91%) with 4-8% monoester P, 3-8% diester P and <1% phosphonates and polyphosphates with no clear trend in relative abundance emerging during the experiment. The NaOH-EDTA extraction resulted in a recovery of ~27% of TP, which is comparable to previous studies with carbonate buffered soils and sediments (Hansen et al., 2004; Turner et al., 2003a). Alpha and beta glycerophosphates are commonly identified in monoester spectral regions and have been demonstrated to be products of diesters degraded during analysis (Doolette et al., 2009; Jørgensen et al., 2015; Paraskova et al., 2015). As no glycerophosphates were identified in any of the analysed samples, recalculation of monoester/diester ratios was not performed. A higher mean monoester/diester ratio (2.31) was found in reduced samples than

Chris Parsons 2017-4-26 19:55
Chris Parsons 2017-4-26 19:55
Chris Parsons 2017-4-26 19:55
Chris Parsons 2017-4-30 12:44
Chris Parsons 2017-4-30 12:43
Chris Parsons 2017-4-30 12:43
Chris Parsons 2017-4-30 12:44
Unknown
Field Code Changed
Chris Parsons 2017-4-30 02:01
Chris Parsons 2017-4-29 16:14
Chris Parsons 2017-4-29 16:14
Chris Parsons 2017-4-30 22:59

[revised manuscript text omitted]